# NKX2-5 mutations causative for congenital heart disease retain functionality and are directed to hundreds of targets

Romaric Bouveret[1,2]*, Ashley J Waardenberg[1,3], Nicole Schonrock[1,2], Mirana Ramialison[1,2,4], Tram Doan[1], Danielle de Jong[1], Antoine Bondue[5,6], Gurpreet Kaur[4], Stephanie Mohamed[1], Hananeh Fonoudi[1,2,7], Chiann-mun Chen[8], Merridee A Wouters[9,10], Shoumo Bhattacharya[8], Nicolas Plachta[4], Sally L Dunwoodie[1,2,11], Gavin Chapman[1,2], Cédric Blanpain[5,12], Richard P Harvey[1,2]

[1]Victor Chang Cardiac Research Institute, Darlinghurst, Australia; [2]St. Vincent's Clinical School, University of New South Wales, Sydney, Australia; [3]Children's Medical Research Institute, Sydney, Australia; [4]European Molecular Biology Laboratory, Australian Regenerative Medicine Institute, Monash University, Clayton, Australia; [5]Institut de Recherche Interdisciplinaire en Biologie Humaine et Moléculaire, Université Libre de Bruxelles, Brussels, Belgium; [6]Department of Cardiology, Erasme Hospital, Brussels, Belgium; [7]Cell Science Research Center, Royan Institute for Stem Cell Biology and Technology, Academic Center for Education, Culture and Research, Tehran, Iran; [8]Department of Cardiovascular Medicine, Wellcome Trust Centre for Human Genetics, University of Oxford, Oxford, United Kingdom; [9]Bioinformatics, Olivia Newton-John Cancer Research Institute, Melbourne, Australia; [10]School of Life and Environmental Sciences, Deakin University, Geelong, Australia; [11]School of Biotechnology and Biomolecular Sciences, University of New South Wales, Kensington, Australia; [12]Walloon Excellence in Life Sciences and Biotechnology, Université Libre de Bruxelles, Brussels, Belgium

*For correspondence:
r.bouveret@victorchang.edu.au

Competing interests: The authors declare that no competing interests exist.

**Abstract** We take a functional genomics approach to congenital heart disease mechanism. We used DamID to establish a robust set of target genes for NKX2-5 wild type and disease associated NKX2-5 mutations to model loss-of-function in gene regulatory networks. NKX2-5 mutants, including those with a crippled homeodomain, bound hundreds of targets including NKX2-5 wild type targets and a unique set of "off-targets", and retained partial functionality. NKXΔHD, which lacks the homeodomain completely, could heterodimerize with NKX2-5 wild type and its cofactors, including E26 transformation-specific (ETS) family members, through a tyrosine-rich homophilic interaction domain (YRD). Off-targets of NKX2-5 mutants, but not those of an NKX2-5 YRD mutant, showed overrepresentation of ETS binding sites and were occupied by ETS proteins, as determined by DamID. Analysis of kernel transcription factor and ETS targets show that ETS proteins are highly embedded within the cardiac gene regulatory network. Our study reveals binding and activities of NKX2-5 mutations on WT target and off-targets, guided by interactions with their normal cardiac and general cofactors, and suggest a novel type of gain-of-function in congenital heart disease.

## Introduction

The mammalian heart is a highly modified muscular vessel whose lineage programmes are governed by conserved gene regulatory networks (GRNs) (*Davidson and Erwin, 2006*). The cardiomyocyte GRN is controlled by lineage-restricted transcription factors (TFs), which interact to form a recursively

**eLife digest** Many genes working within large gene networks influence the development of heart muscle cells in humans and other animals. The activity of these genes is controlled in part by proteins called transcription factors, which bind to DNA and act as molecular switches. One transcription factor that is particularly important for the development of heart muscle cells is called NKX2-5.

Mice lacking NKX2-5 have abnormal hearts and many humans who are born with congenital heart disease carry mutations in the gene that encodes this protein. Many of these mutations alter a section of the protein called the homeodomain, and therefore interfere with the ability of NKX2-5 to bind to DNA or associate with other important cardiac proteins called cofactors. However, it is not clear how such mutations alter the behaviour of NKX2-5 across all of its targets.

Bouveret et al. have now used a technique called 'DNA adenine methyltransferase identification' to study how NKX2-5 interacts with other proteins and DNA. The experiments found that, as expected, the mutant NKX2-5 proteins were unable to associate with many of the usual gene and protein targets of normal NKX2-5. However, the mutant proteins were still able to bind to some of their usual targets, plus many other targets that the normal NKX2-5 protein was not able to bind to.

A particular NKX2-5 mutant protein that the experiments analysed was missing the entire homeodomain, yet it was still able to associate with the normal NKX2-5 protein and bind to cofactors that help NKX2-5 to find its usual targets. This finding led Bouveret et al. to discover the role of a section of the NKX2-5 protein called the tyrosine-rich domain, which in the absence of the homeodomain can direct interactions of NKX2-5 with itself and its cofactors.

Bouveret et al.'s findings suggest that protein cofactors of NKX2-5 help mutant NKX2-5 proteins retain some of their normal activities, but also direct the mutant proteins to abnormal gene targets, which could contribute to congenital heart disease. The next steps are to carry out experiments in animals to confirm these findings, and to understand the activities of mutant NKX2-5 and other mutant transcription factors across the whole genome. This could lead to new therapeutic approaches to treat congenital heart disease and other conditions.

wired sub-network termed the cardiac 'kernel' (*Davidson and Erwin, 2006*). Kernel TFs, such as NKX2-5, GATA4, TBX5/20, and serum response factor (SRF), show regionally restricted expression and act as selector proteins that help define developmental and organ-specific territories. A well-studied kernel TF is NKX2-5, a member of the NK-2 homeodomain (HD) factor subclass that plays an early role in development of hearts and heart-like organs of diverse species (*Elliott et al., 2010*). Loss of *Nkx2-5* in mice leads to arrested heart morphogenesis due to blocked progenitor growth, defective chamber and conduction system development, and a deranged GRN (*Prall et al., 2007*). In humans, NKX2-5 is one of the most commonly mutated single genes in congenital heart disease (CHD), with dominant alleles causing atrial septal defect and atrioventricular conduction block most commonly, and a host of more severe defects at lower penetrance (*Elliott et al., 2010*). Many CHD-causing NKX2-5 mutations are located within the conserved HD (*Figure 1A*), which serves as both a sequence-specific DNA-binding domain and protein-binding interface for interactions with other kernel TFs (*Elliott et al., 2010*). It is widely assumed that CHD is caused by haploinsufficiency and an inability of mutant proteins to recognise target genes. However, we know little about disease causation at the genome level, and in fact, most CHD mutations lie outside of the HD (*Figure 1A*), often in conserved domains with largely unknown functions.

Here, we take a functional genomics approach to understanding the mechanism of NKX2-5 CHD at the chromatin level. We applied the technique of DNA adenine methyltransferase identification (DamID) (*Vogel et al., 2006*) to identify target genes of NKX2-5 wild type and NKX2-5 mutant proteins mimicking those found in patients with CHD. While binding of severe NKX2-5 mutants to targets was compromised, they nonetheless associated with hundreds of targets, including many normally bound by NKX2-5 wild type, and could regulate a subset of these in cellular assays. We demonstrate that severe NKX2-5 mutant proteins retained an ability to interact with NKX2-5 wild type and other cardiac TFs via a novel tyrosine-rich protein:protein interface in NKX2-5 that lies outside of the HD. NKX2-5 mutant proteins also bound hundreds of 'off-targets' not bound by NKX2-5 wild type,

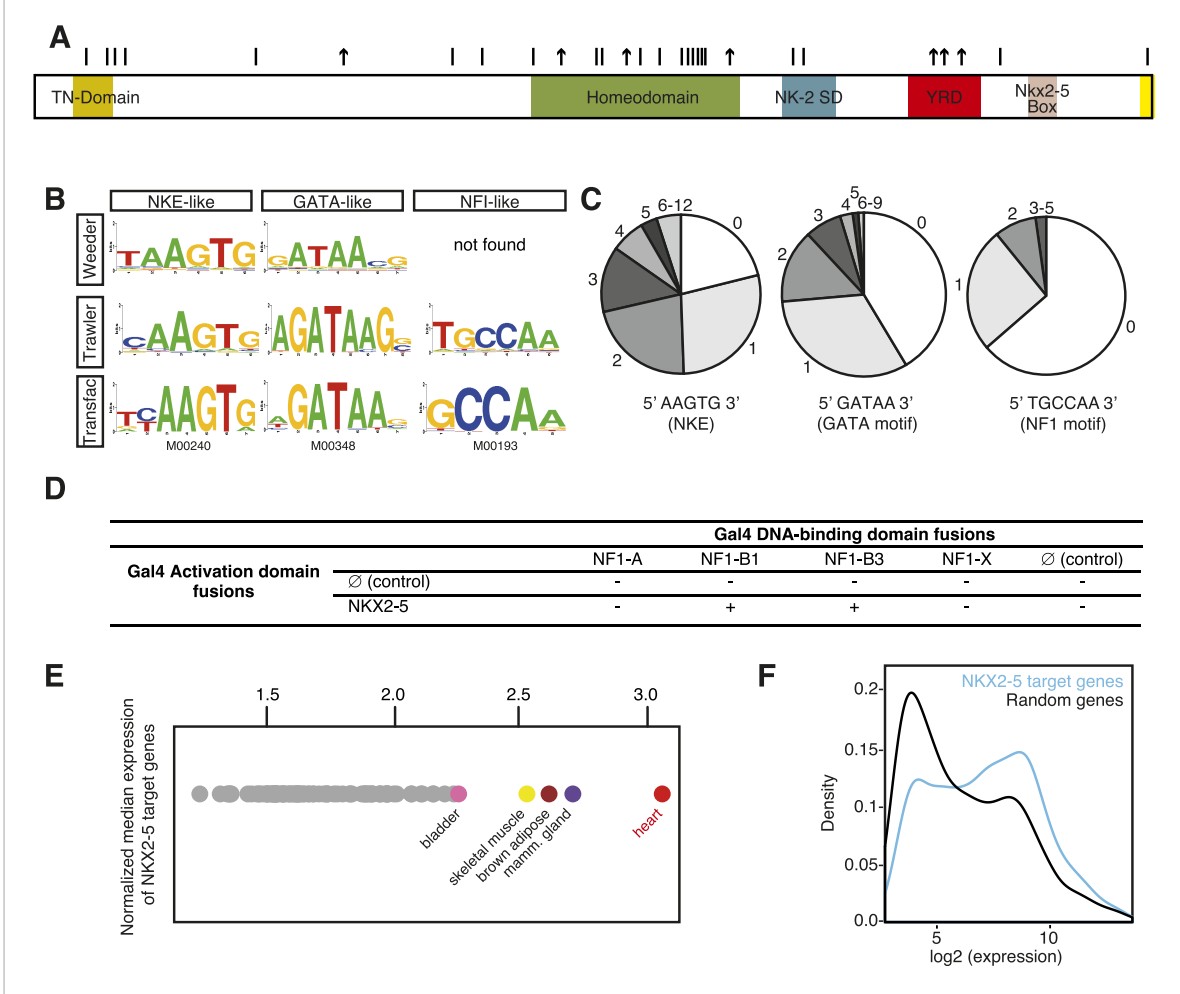

**Figure 1**. DNA adenine methyltransferase identification (DamID) identifies a robust set of NKX2-5 targets in HL-1 cardiomyocytes. (**A**) Structure of the human NKX2-5 protein (TN, tinman domain; NK2SD, NK-2 specific domain; YRD, tyrosine-rich domain). Bars and arrows indicate missense and termination mutations associated with congenital heart disease (CHD), respectively. (**B**) Top over-represented motifs discovered de novo in NKX2-5 peaks using *Trawler* or *Weeder*. NKX2-5, GATA, and Nuclear Factor 1 (NF1) binding motifs deposited in *TRANSFAC* are shown. (**C**) Distribution of NKX2-5, GATA, and NF1 binding sequences in NKX2-5 peaks. (**D**) Yeast-two-hybrid assay. NKX2-5 and NF1 proteins were fused to Gal4-activation and DNA-binding domains, respectively. Positive signs (+) show interaction as growth on selective medium from three independent experiments. (**E**) Normalized median expression of NKX2-5-target genes in 91 murine cell types (data collected from *BioGPS*). Tissues with the highest median expressions are shown in colour, including heart (red). (**F**) Expression of NKX2-5 target genes and random genes in HL-1 cells. Data collected from (*Mace et al., 2009*).

The following figure supplements are available for figure 1:

**Figure supplement 1**. DamID validation in HL-1 cardiomyocytes.

**Figure supplement 2**. Expression of PGK-GFP control 24 hr post-transduction of HL-1 cells transduced with lentivirus (LV).

**Figure supplement 3**. PCR-amplified methylated fragments of HL-1 genomic DNA 40 hr post-transduction with Dam alone and Dam-NKX2-5.

**Figure supplement 4**. False discovery rate of Dam/TF fusion protein binding peaks as determined using CisGenome/TileMapv2 with moving average ≥ 3.5.

**Figure supplement 5**. Chromatin immunoprecipitation (ChIP)-PCR validation of NKX2-5 WT target peakes determined by DamID.

**Figure supplement 6**. NKX2-5 binds the NKE but not the NF1-like motif.

*Figure 1. continued on next page*

*Figure 1. Continued*

**Figure supplement 7**. Identification of NKX2-5, SRF, and ELK1/4 target genes in HL-1 cells.

**Figure supplement 8**. Nuclear localisation of NKX2-5 and histone modifications in HL-1 cardiomyocyte nuclei.

**Figure supplement 9**. Proportional Venn diagram showing the overlapping binding peaks between NKX2-5 determined by DamID (this study) or ChIP-seq (*He et al., 2011*) and (*van den Boogaard et al., 2012*).

via altered DNA-recognition or via interactions with previously unrecognised cofactors. These cofactors, which include members of the ETS family, are embedded in the cardiac GRN, indicating a broad role for ubiquitous TFs in cardiac network logic and mechanism of CHD.

## Results

### DamID identification of NKX2-5 WT targets

DamID is a sensitive enzymatic method developed to identify protein:DNA interactions (*van Steensel and Henikoff, 2000*; *van Steensel et al., 2001*). It is complementary to chromatin immunoprecipitation (ChIP) but avoids artifacts associated with chromatin crosslinking and poor-quality ChIP antibodies (*Waldminghaus and Skarstad, 2010*; *Teytelman et al., 2013*). DamID involves creation of a fusion between a chromatin-binding protein of interest and *Escherichia coli* DNA adenine methyltransferase (Dam). When expressed, Dam fusions bind to DNA, whereby Dam locally methylates adenine within its target sites (5′GATC3′). Target DNA can then be released using *Dpn*I, a nuclease specific for the methylated Dam site, and analysed on arrays (*Vogel et al., 2006*).

Here, we used DamID in HL-1 cells, which resemble atrial cardiomyocytes, to profile NKX2-5 WT and mutants that mimic those found in CHD (*Claycomb et al., 1998*). Mouse *Nkx2-5* cDNAs were cloned into a lentiviral Dam vector carrying heat shock and ponasterone A-inducible promoters (*Vogel et al., 2006*). Only N-terminal Dam fusions (Dam-NKX2-5) were analysed because C-terminal fusions were sterically compromised (data not shown). Ponasterone A induction after viral transduction of HEK Ecr-293 cells confirmed nuclear localisation and the correct size of fusions (*Figure 1—figure supplement 1*). Transduction of HL-1 cells was efficient (~97%; *Figure 1—figure supplement 2*). DamID fusions were expressed from the uninduced *heat shock protein 68* promoter at very low levels (undetectable by western blotting), which was sufficient for Dam enzymatic activity on chromatin (*Figure 1—figure supplement 3*) but avoided skewing of the network. DamID is dependent on the density of *Dpn*I sites (5′GATC3′), which occur on average every 260 bp in the mouse genome, as in the fly genome (*van Steensel and Henikoff, 2000*). After *Dpn*I digestion of virus-transduced HL-1 cell DNA, PCR-amplification lead to fragments of ~200-2000 bp (*Figure 1—figure supplement 3*). We determined that of all perfect NKX2-5 high-affinity binding sites (NKE; 5′AAGTG3′) in the mouse genome, 90% had a *Dpn*I site within 1 kb upstream and downstream (median of ~289 bp), suggesting that DamID captures the vast majority of NKX2-5 direct targets.

Methylated genomic fragments were hybridised to Affymetrix Mouse promoter microarrays in biological triplicate. These microarrays represent 7.3% of the mouse genome, and we estimate from published NKX2-5 ChIPseq data that they would capture 20–30% of all NKX2-5 peaks (*He et al., 2011*; *van den Boogaard et al., 2012*). DamID peaks were determined using *CisGenome* after subtraction of signal obtained from cells expressing Dam alone (*Ji et al., 2008*) (see *Supplementary file 1*, and UCSC genome browser http://genome.ucsc.edu/cgi-bin/hgTracks?db=mm9&type= bed&hgt.customText=ftp://ftp.ncbi.nlm.nih.gov/geo/series/GSE44nnn/GSE44902/suppl/GSE44902_ DamID.bed.gz [*Kent et al., 2002*]).

For NKX2-5 WT, we identified 1524 peaks, which displayed low-false discovery rates (<0.08; *Figure 1—figure supplement 4*). ChIP-PCR confirmed NKX2-5 occupancy over NKX2-5 DamID peaks for 10 out of 11 targets tested, while DamID-negative regions were not occupied (*Figure 1—figure supplement 5*). De novo motif discovery using *Weeder* or *Trawler* (*Pavesi and Pesole, 2006*; *Haudry et al., 2010*) revealed that the top 3 over-represented motifs were 5′AAGTG3′, identical to the NKE;

5′GATAA3′, identical to the GATA-binding motif; and 5′TGCCAA3′, similar to the binding motif of Nuclear Factor 1 (NF1) (*Figure 1B*). After counting of their exact sequences, the NKE was present in 79% of NKX2-5 peaks, with half of these bearing over 2 and up to 12 NKEs. The GATA- and NF1-binding motifs were present in 59% and 36% of peaks, respectively (*Figure 1C*). These results suggest that the majority of NKX2-5 WT targets are directly bound via the NKE and also support the combinatorial action of NKX2-5 with GATA factors on many targets (*Durocher et al., 1997*; *He et al., 2011*). NKX2-5 did not bind directly to the NF1-like site in vitro (*Figure 1—figure supplement 6*), suggesting that NKX2-5 could act in combination with NF1 in HL-1 cells. Using a yeast two-hybrid (Y2H) assay, we showed that NKX2-5 interacted with NF1-B1 and NF1-B3 but not NF1-A or NF1-X (*Figure 1D*), as for the related family members NKX2-1 and NF1 in lung (*Bachurski et al., 2003*). This suggests that NF1-B1/3 regulate a subset of target genes in combination with NKX2-5 in cardiomyocytes.

Using GREAT (*McLean et al., 2010*), NKX2-5 peaks were assigned to 1490 unique genes, including previously identified NKX2-5 target genes (*Figure 1—figure supplement 7*). NKX2-5 also bound its own promoter, potentially reflecting auto-regulation (*Prall et al., 2007*). Gene Ontology (GO) analysis of DamID NKX2-5 targets showed that the most enriched biological processes were *heart development* and *muscle contraction* (*Supplementary file 1*), attesting to the specificity of DamID. The most represented words amongst the top 50 GO terms, were *muscle*, *development*, *regulation*, *cardiac*, and *cell*. Other enriched processes included *cytoskeleton organisation* and related terms, and metabolic GO terms such as *regulation of cellular carbohydrate catabolic process*, *regulation of glycolysis*, and *regulation of generation of precursor metabolites and energy*, indicating that NKX2-5 exerts high-level network control over metabolic processes.

In silico analysis using the *BioGPS* repository of 91 tissue transcriptomes revealed that the median expression level of NKX2-5 target genes was highest in adult hearts (*Figure 1E*). Transcriptome data for HL-1 cells (*Mace et al., 2009*) showed that NKX2-5 targets were strongly skewed towards higher expression values compared to genes randomly selected from the array, although some genes were expressed at lower levels (*Figure 1F*). We note from immunofluorescence studies that the most intense NKX2-5 staining co-localised in speckle-like nuclear foci with active histone marks H3K9Ac and H3K4me3, but not with repressive mark H3K27me3 (*Figure 1—figure supplement 8*).

## Probing the mechanisms of CHD using DamID

Having established a robust NKX2-5 WT target list, we set out to probe the mechanisms of CHD by identifying the genome-wide targets of NKX2-5 mutant proteins. We first constructed N-terminal Dam fusions for the human CHD mutation NKX2-5Y191C (*Benson et al., 1999*), which disrupts tyrosine 54 of the HD, necessary for the unique binding site specificity of the NK-2 homeoprotein sub-class to the NKE (*Figure 2A*). In vitro, NKX2-5Y191C dimerises normally with co-factors but shows 80-fold reduced binding to the NKE (*Kasahara et al., 2000*; *Kasahara and Benson, 2004*). We also created NKX2-5ΔHD, in which the HD was deleted and replaced by a glycine-linker. We anticipated that NKX2-5ΔHD would be functionally dead, since the HD has been shown to serve as both DNA-binding domain and interface for homo- and hetero-dimerisation (*Elliott et al., 2010*). However, we were interested in testing whether the presence of other conserved domains in NKX2-5 and often mutated in NKX2-5-related CHD (*Figure 2A*) could confer any functionality to the NKX2-5 mutants. Both mutants were stable (*Figure 1—figure supplements 1, 6*) and, as anticipated, did not bind efficiently to the NKE of the known target *Nppa* (natriuretic peptide precursor A) (*Durocher et al., 1996*) in vitro (*Figure 1—figure supplement 6*).

Surprisingly, in HL-1 cells, both NKX2-5Y191C and NKX2-5ΔHD associated with a large number of loci (1149 and 792 peaks, respectively; *Figure 2B*; *Supplementary file 1*) with low false discovery rates (*Figure 1—figure supplement 4*). Even though NKX2-5Y191C cannot bind DNA, the most over-represented motifs were 5′AAGTGT3′ (NKE), 5′GATAA3′ (GATA), and 5′TGCCAA3′ (NF1-like), exactly as for NKX2-5 WT fusion (*Supplementary file 1*). NKX2-5Y191C peaks also showed over-representation of the motif 5′TAATC3′, which is similar to the binding sites of many non-NK-2 class HD proteins, including HOX proteins, as well as that of NK-2 proteins that lack HD tyrosine 54, such as NKX1-2 (*Berger et al., 2008*). NK-2 class proteins which do carry Y54 in their HDs, including NKX2-5, also bind to this HOX-like site, albeit with a 10-fold-reduced affinity compared to that of the NKE (*Chen and Schwartz, 1995*). For NKX2-5ΔHD peaks, there were many over-represented motifs, although none resembled known cardiac TF-binding sites (*Supplementary file 1*).

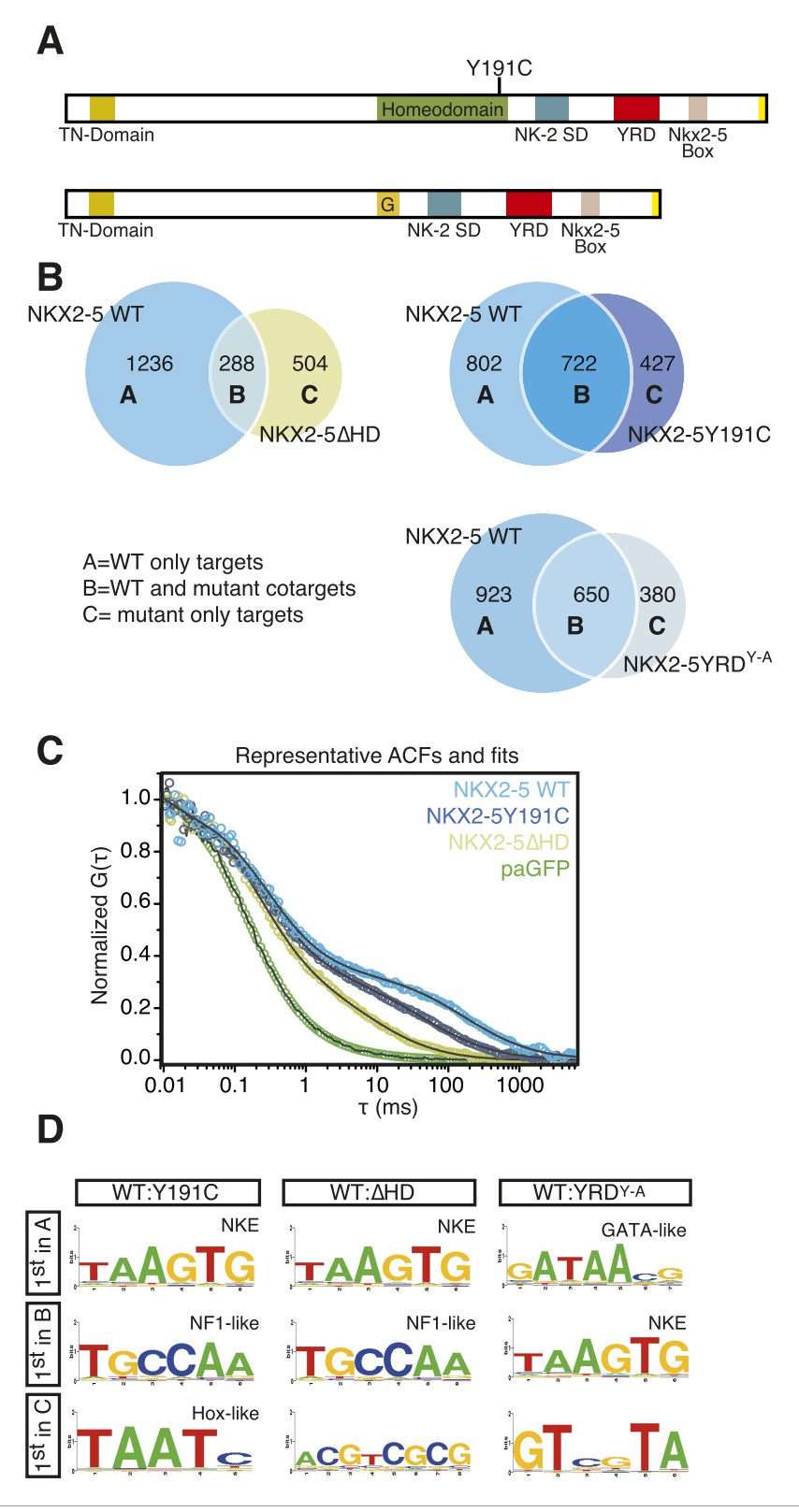

**Figure 2**. NKX2-5 mutants bind to hundreds of targets in HL-1 cells. (**A**) Structure of NKX2-5Y191C and NKX2-5ΔHD mutant proteins ('G' indicates a Glycine linker). (**B**) Overlapping binding peaks between NKX2-5 WT and mutants. Proportional Venn diagrams show peaks unique to WT NKX2-5 (A sets); common to WT and mutant proteins (B sets);

*Figure 2. continued on next page*

*Figure 2. Continued*

unique to mutant proteins (C sets). (**C**) Representative autocorrelation function (ACF) curves and fits for NKX2-5 WT and mutant proteins measured by photoactivatable fluorescence correlation spectroscopy (paFCS) in HL-1 cells. (**D**) Top over-represented motifs discovered de novo using *Weeder* in A-, B- and C-target sets of NKX2-5 WT and mutant comparisons.

The following figure supplements are available for figure 2:

**Figure supplement 1**. (**A**) Image showing paGFP-NKX2-5 and histone 2B-RFP fluorescence in a single HL-1 nucleus with fluorescence correlation spectroscopy (FCS) measurement performed at the crosshair point.

**Figure supplement 2**. Enrichment of known motifs in NKX2-5 peak subsets.

**Figure supplement 3**. Enrichment of GO terms in NKX2-5 target subsets.

---

The ability of NKX2-5Y191C and NKX2-5ΔHD to bind a large number of targets was unexpected. Hence we sought to confirm using an independent approach that these mutants could interact with chromatin in vivo. We employed photoactivatable fluorescence correlation spectroscopy (paFCS) (*Kaur et al., 2013*), measuring the diffusion dynamics of NKX2-5 WT and mutants within individual HL-1 nuclei (*Figure 2—figure supplement 1A*). Expression vectors encoding NKX2-5 proteins fused to photoactivatable GFP (paGFP) were introduced by transfection. As for other TFs, NKX2-5 WT showed a biphasic behaviour, with a freely diffusing fraction similar to paGFP (adjusted for molecular mass) and a slower, chromatin-interacting fraction. NKX2-5 WT displayed slower diffusion (dwell time $\tau D_{slow}$ = 121 ms) than that mutants (NKX2-5Y191C: $\tau D_{slow}$ = 64 ms; NKX2-5ΔHD: $\tau D_{slow}$ = 11 ms), which diffused more slowly than paGFP alone (*Figure 2C*, *Figure 2—figure supplement 1B-D*). This confirmed that NKX2-5Y191C and NKX2-5ΔHD physically interact with chromatin (*Kaur et al., 2013*) and suggests that a gradation in chromatin tethering correlates with the apparent severity of the NKX2-5 mutation (WT>Y191C>ΔHD>paGFP).

## NKX2-5 mutant targets overlap partially with NKX2-5 WT targets

We compared NKX2-5 WT and mutant peaks and adopted a simple nomenclature (A, B, and C) for peak subsets, where A is bound by NKX2-5 WT only, B is bound by both WT and a mutant, and C is bound only by a mutant (*Figure 2B*). In both comparisons, the A sets predominated (81% and 53% of NKX2-5 WT peaks in the NKX2-5 WT/ΔHD and NKX2-5 WT/Y191C comparisons, respectively), and their most enriched motif determined de novo was the NKE (*Figure 2D*), demonstrating that, as anticipated, NKX2-5 mutants failed to bind to most WT targets, including *Nppa*. Therefore, the HD mutations result in a severe loss of function. However, a high proportion of mutant peaks overlapped with NKX2-5 WT peaks (B sets; 36% of 792 NKX2-5ΔHD peaks and 63% of 1149 NKX2-5Y191C peaks).

To investigate the A- and B-set logic further, we used *Clover* (*Frith et al., 2004*) to calculate the enrichment of known TF motifs present in *TRANSFAC* (*BIOBASE*) and *JASPAR* databases (*Bryne et al., 2008*) in A and B sets separately (*Figure 2—figure supplement 2*). The NKE was significantly enriched in both A and B sets, showing that A and B sets contain direct NKX2-5 targets. A- and B-sets also exhibited distinct enrichment signatures for many known TF motifs, for example, those for cardiac kernel TFs GATA4, HAND1, and MEF2 were only enriched in the B-set of the NKX2-5 WT/ΔHD comparison. Thus, both mutants can bind a subset of direct WT targets despite crippled DNA binding. This likely occurs via dimerisation with endogenous NKX2-5 WT and/or its cofactors (see below and [*Kasahara et al., 2000*]). Indeed, for NKX2-5ΔHD, which completely lacks a DNA-binding domain, our data suggest that a protein:protein interface within NKX2-5 that is distinct from the HD can guide the mutant protein to NKX2-5 WT targets.

At a GO term level, the A- and B-sets were also distinct (*Figure 2—figure supplement 3*; *Supplementary file 1*). For the Y191C/ΔHD comparison, the A-set showed virtually no enrichment for GO terms and the B-set was primarily enriched in functions related to *heart development*. For the WT/ΔHD comparison, the overlap in GO terms between the A-set and B-set was also very small, with the A-set being primarily enriched in terms related to cellular functions, such as *actin filament*

*organisation* and *transcription*, and the B-set being enriched in functions related to *heart* and *blood vessel development*. This suggests that NKX2-5 mutants preferentially associate with genes involved in cardiac related processes.

## Functionality of NKX2-5 mutants

Based on the DamID and pFCA data above showing an association between NKX2-5 mutants and chromatin, we tested the functionality of NKX2-5 mutants in vivo, selecting *Inhibitor of DNA binding 3* (*Id3*) as a B set gene bound by NKX2-5 WT, Y191C, and ΔHD for detailed analysis (*Figure 3A*). *Id3* encodes a bHLH factor that represses cardiogenic differentiation in heart progenitor cells and is down-regulated as progenitors differentiate and levels of NKX2-5 increase (*Ding et al., 2006*). Using qRT-PCR, we confirmed previous microarray data that *Id3* is up-regulated in *Nkx2-5* null mouse mutant hearts, similar to other cardiac progenitor genes (*Prall et al., 2007*), whereas *Nppa*, a known directly activated NKX2-5 target, is down-regulated (*Figure 3—figure supplement 1*). The *Id3* downstream region, when cloned into a minimal promoter/luciferase vector in antisense orientation and transfected into HEK 293 cells, stimulated transcription ~18-fold. Activity was stimulated a further fourfold by SRF (*Figure 3—figure supplement 1*), which DamID indicated also bound to this region (*Figure 3A* and below). NKX2-5 WT, Y191C, and ΔHD repressed both the basal and SRF-stimulated activity (*Figure 3B*), suggesting that NKX2-5 mutants retain some WT functionality as repressors of *Id3*.

To test functionality of NKX2-5 mutants in a developmental context, we generated mouse embryonic stem (ES) cells in which expression of exogenous NKX2-5 WT or mutants could be conditionally induced by doxycycline (dox)-dependent activation of a ubiquitous promoter (*Bondue et al., 2008, 2011*). Following induction of cardiogenesis, NKX2-5 WT, Y191C, and ΔHD proteins were activated in most or all embryoid body (EB) cells with dox at day 4, 1 day before, endogenous NKX2-5 appears in a minority of cells within cardiogenic clusters (*Figure 3C*). Comparable levels of NKX2-5 WT and mutant mRNAs and protein were detected 1–2 days post-induction (*Figure 3—figure supplement 2*). Induced NKX2-5 WT strongly repressed formation of FLK1+/PDGFRα+ multipotent cardiovascular progenitors at day 5 (*Figure 3D*), and of cardiomyocytes (cTNT+) and endothelial cells (CD31+) at day 8 (*Figure 3E-F*). Repression of cardiac lineages by NKX2-5 in ES cells is consistent with its demonstrated early role as a negative feedback regulator of cardiac induction and second heart field gene expression (*Prall et al., 2007*), and as an inhibitor of reprogramming of cardiac fibroblasts to a cardiomyocyte fate (*Ieda et al., 2010*). NKX25ΔHD was nuclear in only 15% of cells at day 6 (likely due to deletion of the nuclear localisation signal within the HD; *Figure 3—figure supplement 2C*) and had no effect on cardiac or endothelial cell differentiation (*Figure 3E-F*) and so was not considered further. NKX2-5Y191C was nuclear and, while showing no effects on myocardial progenitors or endothelial cells in three independent experiments, showed a trend towards inhibition of cardiomyocyte numbers at day 8 (~44%; p = 0.079), suggesting that it also retained some repressive activity in this assay.

To gain deeper insights into the underlying effects, we therefore measured alterations in gene expression in the ES cell lines at day 5–6 in three independent experiments, testing a selection of A and B-set genes derived from the NKX2-5WT:Y191C comparison. The majority of both A (8/12) and B (12/13) set genes tested were modulated >1.5 fold by NKX2-5 WT (total 20/25), providing strong evidence that DamID selects NKX2-5 WT developmental targets. Within B-set genes, most were repressed by NKX2-5 WT, consistent with the repression of cardiogenesis by NKX2-5 WT at early stages of cardiac lineage specification (*Prall et al., 2007*). A subset of A and B set genes was also regulated by NKX2-5Y191C (A-set: 3/12; B-set: 4/13; *Figure 3G*), despite no change in early myocardial progenitors in the NKX2-5Y191C line, with regulation by Y191C being in the same direction as for NKX2-5 WT. As examples, *Actc1*, *Nkx2-5*, *Hand2*, and *Gata4* were repressed by both NKX2-5 WT and NKX2-5Y191C, while *Nppa* and *Tbx3* were activated by both. These data show that while NKX2-5Y191C loses its ability to bind and regulate most normal NKX2-5 targets, it nonetheless retains some of the regulatory capabilities of NKX2-5 WT.

## Novel interface for NKX2-5 dimerisation

Our results showing that the NKE is enriched in both B-sets suggest that NKX2-5 mutants can bind targets by dimerising with NKX2-5 WT. While dimerisation to NKX2-5 WT has been demonstrated for NKX2-5Y191C (*Kasahara and Benson, 2004*), it was surprising for NKX2-5ΔHD because the HD has

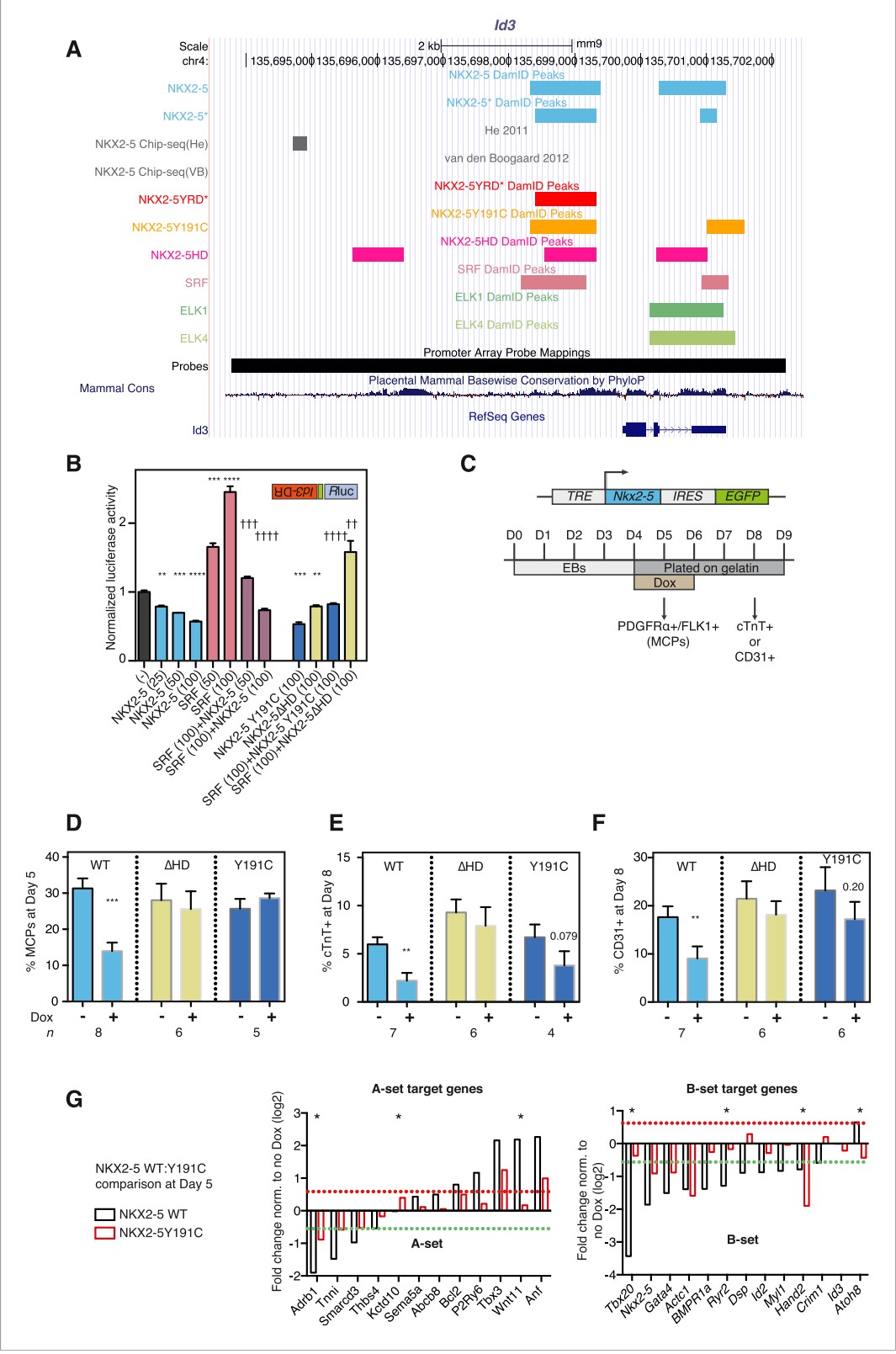

**Figure 3**. NKX2-5 mutant proteins retained partial functionality. (**A**) UCSC Genome Browser screen shot showing DamID transcription factor (TF) association in HL-1 cells with *Inhibitor of DNA binding 3* (*Id3*). (**B**) Normalised *R*luc activity in HEK 293 cells transfected with NKX2-5 and serum response factor (SRF). *Id3 DR* was in the antisense orientation. p-values were calculated relative to controls (** p < 0.01; *Figure 3. continued on next page*

*Figure 3. Continued*

\*\*\* p < 0.001; \*\*\*\* p < 0.0001; ns = not significant) or SRF alone (†† p < 0.01; ††† p < 0.001; †††† p < 0.0001). The quantity of vector is given in ng. (**C**) Schematic representation of 'Materials and methods' for the generation of embryonic stem (ES) cell lines for inducible expression of NKX2-5 WT and mutants. Cardiac differentiation was initiated in embryoid bodies (EBs), followed by induction of cardiogenesis differentiation (from day 4) on plates. (**D**) FACS quantification of PDGFRa/FLK1+ multipotent cardiovascular progenitors (MCPs) in NKX2-5 dox-inducible ES cell lines (at day 5). The mean of 6–7 independent experiments is shown. (**E, F**) FACS quantification of cTNT + cardiomyocytes (**D**) and CD31+ endothelial cells (**E**) at day 8. (**G**) RT-PCR quantification of A- and B-set target genes in NKX2-5 WT or Y191C dox-inducible ES cell lines at day 5 in three independent experiments (24 hr post-induction). Results were normalised to expression in uninduced cells. p-values were calculated using a t-test (\* p < 0.05).

The following figure supplements are available for figure 3:

**Figure supplement 1**. WT and mutant NKX2-5 targets identified by DamID in HL-1 cells.

**Figure supplement 2**. (**A, B**) RT-PCR quantification of *Nkx2-5* ORF and endogenous *Nkx2-5* (specifically detected using primers in the *Nkx2-5* 3′UTR, which is absent in the inducible construct) in NKX2-5 WT/Y191C/ΔHD dox-inducible ES cell lines at day 6 (WT:ΔHD comparison (**A**); 48 hr post-dox induction) and day 5 (WT:Y191C comparison (**B**); 24 hr post-induction).

been previously described as being essential for NKX2-5 homodimerisation (*Elliott et al., 2010*). Transfected NKX2-5ΔHD was largely cytoplasmic in CV-1 cells (*Figure 4A*), which lack endogenous NKX2-5. This is likely a consequence of the deletion of the NKX2-5 nuclear localisation signal located in the amino terminus of the HD (*Kasahara and Izumo, 1999*). However, NKX2-5ΔHD became nuclear when co-expressed with NKX2-5 WT, which itself was nuclear when transfected alone (*Figure 4A*). Cofactors GATA4, TBX5, and TBX20, which have been reported to bind NKX2-5 WT via the HD also induced nuclear translocation of NKX2-5ΔHD (*Figure 4—figure supplement 1*). In HL-1 cells, which express endogenous NKX2-5 and its cardiac cofactors, transfected NKX2-5ΔHD was predominantly nuclear although some protein was present in the cytoplasm (*Figure 4B*). These data demonstrate that NKX2-5 WT and its cofactors are able to carry NKX2-5ΔHD into the nucleus via an unknown protein:protein interaction not involving the HD.

To demonstrate the presence of this interface independently, we used a *Renilla* luciferase protein fragment complementation assay (*R*luc-PCA) (*Stefan et al., 2007*) with a weakened Cytomegalovirus (CMV) promoter to avoid protein aggregation. This assay confirmed NKX2-5 WT homo-dimerisation, hetero-dimerisation between NKX2-5 WT and NKX2-5ΔHD, and homo-dimerisation of NKX2-5ΔHD (*Figure 4C*). It also confirmed interactions between NKX2-5ΔHD and GATA4, TBX5, and TBX20 (*Figure 4—figure supplement 2*). We further employed processed spectral Fluorescence Resonance Energy Transfer (psFRET) (*Chen et al., 2007*), detecting robust Fluorescence Resonance Energy Transfer (FRET) from NKX2-5 WT/WT homodimers or WT/ΔHD heterodimers in the nucleus and from ΔHD/ΔHD homodimers in the cytoplasm (*Figure 4D–E*), the latter result demonstrating that dimerisation was independent of chromatin binding.

To define the dimerisation domain, we used the Y2H assay (*Figure 4F*). Full-length NKX2-5 interacted with peptides carrying the NKX2-5 HD (aa137-196) or C-terminal region (aa198-318) but not with the N-terminal region (aa1-135). The C-terminal peptide could also interact with itself. The conserved tyrosine-rich domain (YRD), present within the C-terminal region (*Figure 1A*), was essential for the interaction between full-length NKX2-5 and the C-terminus, being blocked by the NKX2-5YRD$^{Y-A}$ mutation in which all 9 tyrosines of the YRD are mutated to alanine. We have previously shown that NKX2-5YRD$^{Y-A}$ does not affect DNA binding, although it has a strong dominant-negative activity leading to lethal CHD-like phenotypes in high-level heterozygous ES cell chimaeras, and when placed over the null allele creates a phenocopy of NKX2-5 loss-of-function (*Elliott et al., 2006*). Mutation of other conserved domains within the C-terminus (NK2-specific domain; Nkx2-5 box) had no affect on the interaction (*Figure 4F*). While the NKX2-5 HD is known to be a homophilic interaction domain (*Elliott et al., 2010*), our Y2H data identify the YRD as a novel dimerisation interface within NKX2-5. This YRD-dependent interaction, demonstrated in both yeast and mammalian cells, provides the mechanism for heterodimerisation of NKX2-5 WT with NKX2-5ΔHD, as well as homodimerisation of NKX2-5ΔHD.

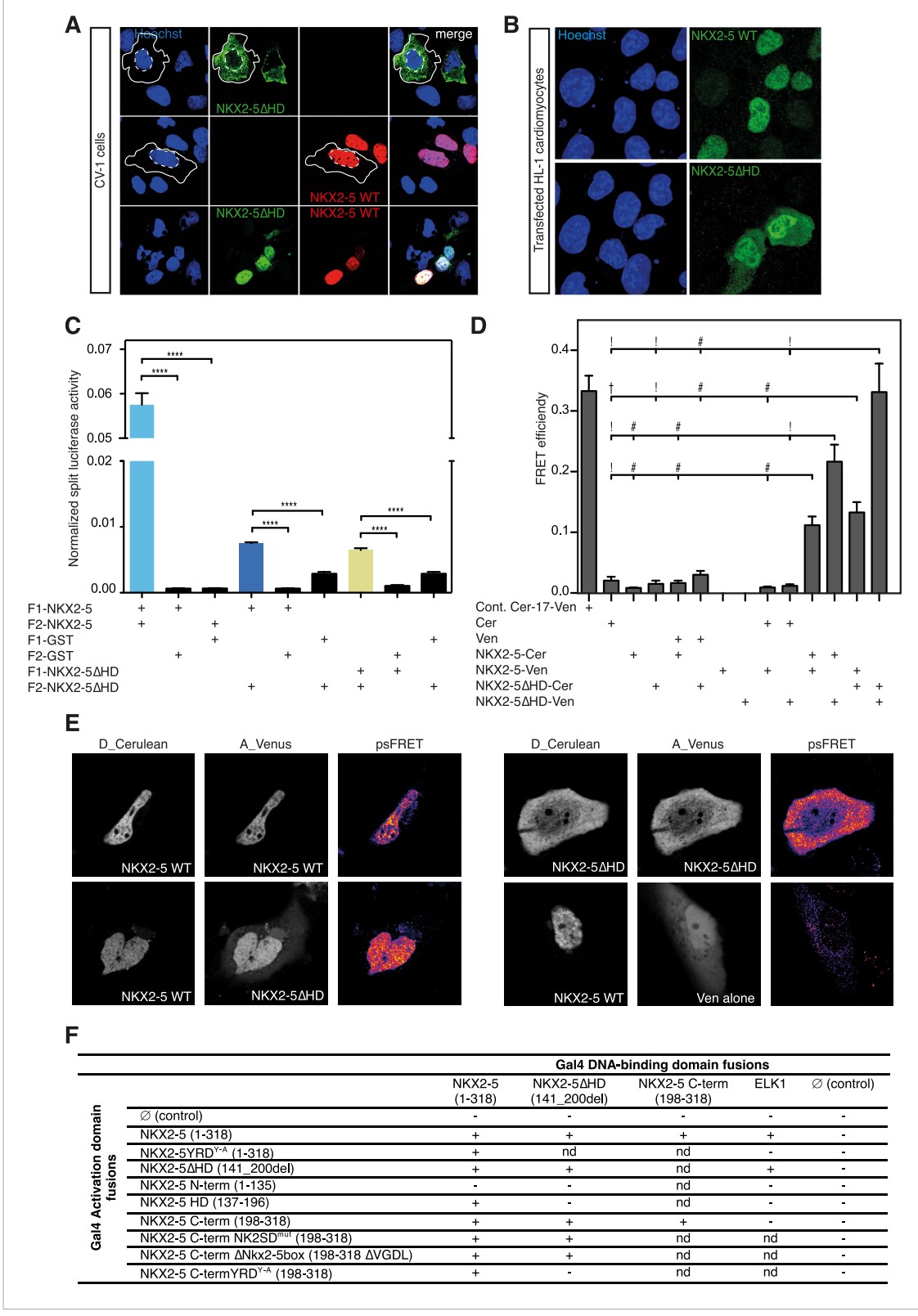

**Figure 4.** The YRD is essential for interaction between NKX2-5 and NKX2-5ΔHD. (**A**) Intracellular localisation of V5-tagged NKX2-5ΔHD and HA-tagged NKX2-5 WT. CV-1 cells were transfected with NKX2-5ΔHD only (top row), NKX2-5 WT only (middle row), or both (bottom row). Solid and dashed lines highlight the cellular and nuclear boundaries, respectively. (**B**) Intracellular localisation of transfected V5-tagged NKX2-5 WT (top row) and NKX2-5ΔHD (bottom row) in HL-1 cells. (**C**) NKX2-5 homo- and hetero-dimerisation measured by *R*luc-PCA in HEK 293T cells. F1 and F2 represent the N- and

*Figure 4. continued on next page*

*Figure 4. Continued*

C-terminal *R*luc fragments, respectively. Data are represented as mean of the normalised luciferase activity ± SEM. Significance was calculated using an unpaired t-test (**** p < 0.0001). (**D**) NKX2-5 homo- and hetero-dimerisation measured by processed spectral Förster Resonance Energy Transfer (psFRET). Cerulean and Venus represent the donor and acceptor molecules. FRET efficiency is represented as mean ± SEM. p-values were calculated using a t-test between each pair and its appropriate controls. Significance is as follows: † for p-value < 0.05; ! for p-value < 0.001 ; # for p-value < 0.0001; values are given if p-value >0.05. (**E**) Representative psFRET images (false coloured using fire look up table) used in (**D**). (**F**) Yeast-two-hybrid assay. Proteins were fused to Gal4-activation and DNA-binding domains. Positive signs (+) show interaction as growth on selective medium from three independent experiments (nd = not determined). FRET, Förster Resonance Energy Transfer.

The following figure supplements are available for figure 4:

**Figure supplement 1**. NKX2-5ΔHD interacts with NKX2-5 cardiac cofactors and ETS-factors in vivo.

**Figure supplement 2**. NKX2-5 dimerisation with TBX5, TBX20, and GATA4 measured by *R*luc-PCA.

**Figure supplement 3**. Density of NKX2-5 WT, NKX2-5ΔHD, and NKX2-5Y191C peak sets and of probes relative to the TSS.

## NKX2-5 mutants bind to off-targets

When comparing NKX2-5 WT and mutant peaks, the presence of C sets revealed that mutant proteins bound a set of unique targets (*Figure 2B*) that we call 'off-targets' potentially reflecting a gain-of-function. Proportionally, off-target sets were much larger for the more severe NKX2-5ΔHD than for NKX2-5Y191C (64% of total NKX2-5ΔHD targets [504 peaks]; 37% of NKX2-5Y191C targets [427 peaks]).

Off-target peaks had specific characteristics. For example, for both NKX2-5ΔHD and Y191C mutants, C-sets peaks were uni-modal centred at the transcription start site, while A- and B-set peaks displayed a bimodal distribution around the transcription start site, as for total NKX2-5 WT peaks (*Figure 4—figure supplement 3*). Testing NKX2-5 mutant off-target genes for over-representation of GO terms, we found that both off-target sets were enriched in GO terms *chromatin organisation*, *macromolecular complex assembly*, and *cell cycle*, but not cardiac terms (*Figure 2—figure supplement 3*). These results indicated that off-targets are selected via a specific logic, most likely via interaction with non-cardiac-restricted cofactors.

## ETS factor binding sites are enriched in NKX2-5ΔHD off-targets

Using Clover (*Frith et al., 2004*) to calculate the enrichment of all known TF motifs present in both C sets (Figure 2—figure supplement 4), we noted that off-target peak sets for NKX2-5ΔHD and Y191C were not enriched in the NKE, nor in fact DNA-binding sites of virtually all other known cardiac TFs (Figure 2—figure supplement 4). The most over-represented motif for NKX2-5Y191C was 5′TAAT3′ (*Figure 2D*), which is similar to the HOX-like site over-represented in NKX2-5 WT targets mentioned above. Our DamID findings on NKX2-5Y191C are consistent with in vitro data showing that the affinity of an analogous *Drosophila* NK2 class HD mutant (vnd/NK2 Y54M) for the NKE is reduced by 10-fold, while its affinity for the TAAT core was unchanged (*Weiler et al., 1998*).

For both mutants, off-target peak sets were enriched in other motifs specific to many broadly expressed TFs that are not known to be part of the cardiac GRN (Figure 2—figure supplement 4). We focused on the enrichment of TF-binding sites in the C-set of NKX2-5ΔHD, based on the hypothesis that DNA-binding cofactors of NKX2-5 guide NKX2-5ΔHD to its off-targets. The motifs for E−26 transformation-specific (ETS) TFs occurred most frequently in NKX2-5ΔHD C-set peaks (42%) and were significantly enriched compared to random peaks, suggesting that ETS factors could play a role in guiding NKX2-5ΔHD to its off-targets. ETS factors are broadly expressed TFs that are activated by phosphorylation downstream of tyrosine kinase receptors. They are important for extracellular signal-gating in specification of cardiac progenitors in *Ciona* (*Davidson et al., 2006*), eve-positive extra-cardiac progenitors in *Drosophila* (*Halfon et al., 2000*), and for specification of vascular endothelial cells and *Gata4* expression in endocardial cushions in mammals (*Schachterle et al., 2011*). However, the broader roles of ETS factors in mammalian cardiogenesis are uncharted.

## ELK1/4 and other ETS factors interact with NKX2-5

Many ETS factors are expressed in embryonic hearts (*Schachterle et al., 2011*), and we selected ternary complex factors ELK1 and ELK4/SAP-1/TCF-1, whose motifs occurred in 26% of C-set peaks, for further analysis. We speculated that these ubiquitous TFs are cofactors of NKX2-5 and, in the absence of compelling cardiac specificity conferred by the HD, guide NKX2-5ΔHD to a subset of off-targets. In the next set of experiments, we set out to establish this principle.

Antibodies specific for total ELK1 or its phosphorylated form (pELK1) showed that in HL-1 cells ELK1 was mostly cytoplasmic and unphosphorylated, although in a minority of cells ELK1 was phosphorylated and nuclear, where it co-localised with endogenous NKX2-5 (*Figure 5A,B*). In CV-1 cells, transfected ELK1 was also variably partitioned between the nucleus and cytoplasm as in transfected neurons (*Lavaur et al., 2007*). When phosphorylated and nuclear, pELK1 guided co-transfected NKX2-5ΔHD into the nucleus of CV-1 cells. Conversely, when unphosphorylated, it retained co-expressed NKX2-5 WT within cytoplasmic inclusions (*Figure 5C*; *Figure 5—figure supplement 1*). These data suggest that ELKs and NKX2-5 are interacting cofactors.

Using *R*luc-PCA in HEK 293 cells, we confirmed that both ELK1 and ELK4 interacted with NKX2-5 WT and NKX2-5ΔHD, as well as SRF and TBX5 (*Figure 5D*, *Figure 5—figure supplement 2*). NKX2-5 WT also interacted with ETS-family members ELK3, ETS1, ERF, and GABPα (*Figure 5—figure supplement 2*). psFRET confirmed the NKX2-5/ELK1 interaction in both the nucleus and cytoplasm in CV-1 cells (*Figure 5E–F*), demonstrating that the interaction was independent of ELK1 phosphorylation and chromatin binding. SRF is a well-described cofactor of both NKX2-5 and ELK1, albeit in different contexts (*Treisman, 1994*). SRF is an effector of several signalling pathways and plays a specific role during early heart formation via its combinatorial action with NKX2-5 and GATA4 on target genes. To test whether the NKX2-5/ELK1 interaction was mediated by SRF, we utilised *R*luc-PCA. The ELK1 B-box mutation Y159A, which disrupts the interaction between ELK1 and SRF (*Ling et al., 1998*), still interacted with NKX2-5 (*Figure 5D*) and, as for ELK1 WT, could induce nuclear translocation of NKX2-5ΔHD or retain NKX2-5 WT in cytoplasmic inclusions (*Figure 5—figure supplement 1*). Thus, the ELK/NKX2-5 interaction does not require SRF.

In the Y2H assay, we confirmed an interaction between ELK1 and both NKX2-5 WT and NKX2-5ΔHD (*Figure 4F*). ELK1 did not interact with the HD, and its interaction with NKX2-5 was blocked by the YRD$^{Y-A}$ mutation, demonstrating that the YRD, in addition to being a haemophilic interaction domain contributing to NKX2-5 homodimerisation, is a protein:protein interface essential for the interaction between NKX2-5 and ELK1. Our results are consistent with previous data showing that a tyrosine-rich region within NK2 class homeo-protein NKX3-1 mediates interaction with prostate-derived ETS factor (*Chen and Bieberich, 2005*). The data suggest that ETS factors including ELK1/4 are functional cofactors of NKX2-5 WT during cardiogenesis.

## Identification of ELK1/4 targets in HL-1 cells

In order to confirm binding of ELK1/4 to NKX2-5ΔHD off-targets, we identified ELK1 and ELK4 target peaks in HL-1 cells using DamID. For ELK1 and ELK4, we generated robust data for both N- and C-terminal Dam fusions. All data sets showed low false discovery rates (*Figure 1—figure supplement 4*) and significant overlap between N- and C-terminal Dam fusions (p < 0.001), resulting in 1217 and 875 overlapping peaks for ELK1 and ELK4 fusions, respectively (*Figure 6A*; *Supplementary file 1*). De novo motif discovery identified the known DNA-binding site for both ELK1 and ELK4 as the only over-represented motif (*Figure 6B*; *Supplementary file 1*). The overlap between ELK1 and ELK4 peaks (p < 0.001) was high (48% of ELK1 and 67% of ELK4 peaks; *Figure 6A*).

ELK1/4 peaks could be assigned to 1423 and 1051 unique target genes, respectively. GO analysis showed highest over-representation of terms *cytoskeletal organisation* and *RNA processing* (*Supplementary file 1*), suggesting that, overall, ELK1/4 regulate many generic cellular functions. ELK1/4 target gene median expression was low in heart and highest in mast cells and macrophages (*Figure 6C*). In contrast to NKX2-5 WT peaks, ELK1/4 peaks showed unimodal enrichment centred across the transcription start site (TSS), identical to those of NKX2-5ΔHD off-targets (*Figure 6D,S6C*). These data highlight the architectural differences between the majority of NKX2-5 WT peaks and those bound by ELK1/4 and NKX2-5ΔHD.

Having established a robust target set for ELK1/4, we determined how many peaks within the NKX2-5ΔHD C-set were actually occupied by ELK1/4 in HL-1 cells. Of the 506 NKX2-5ΔHD C-set

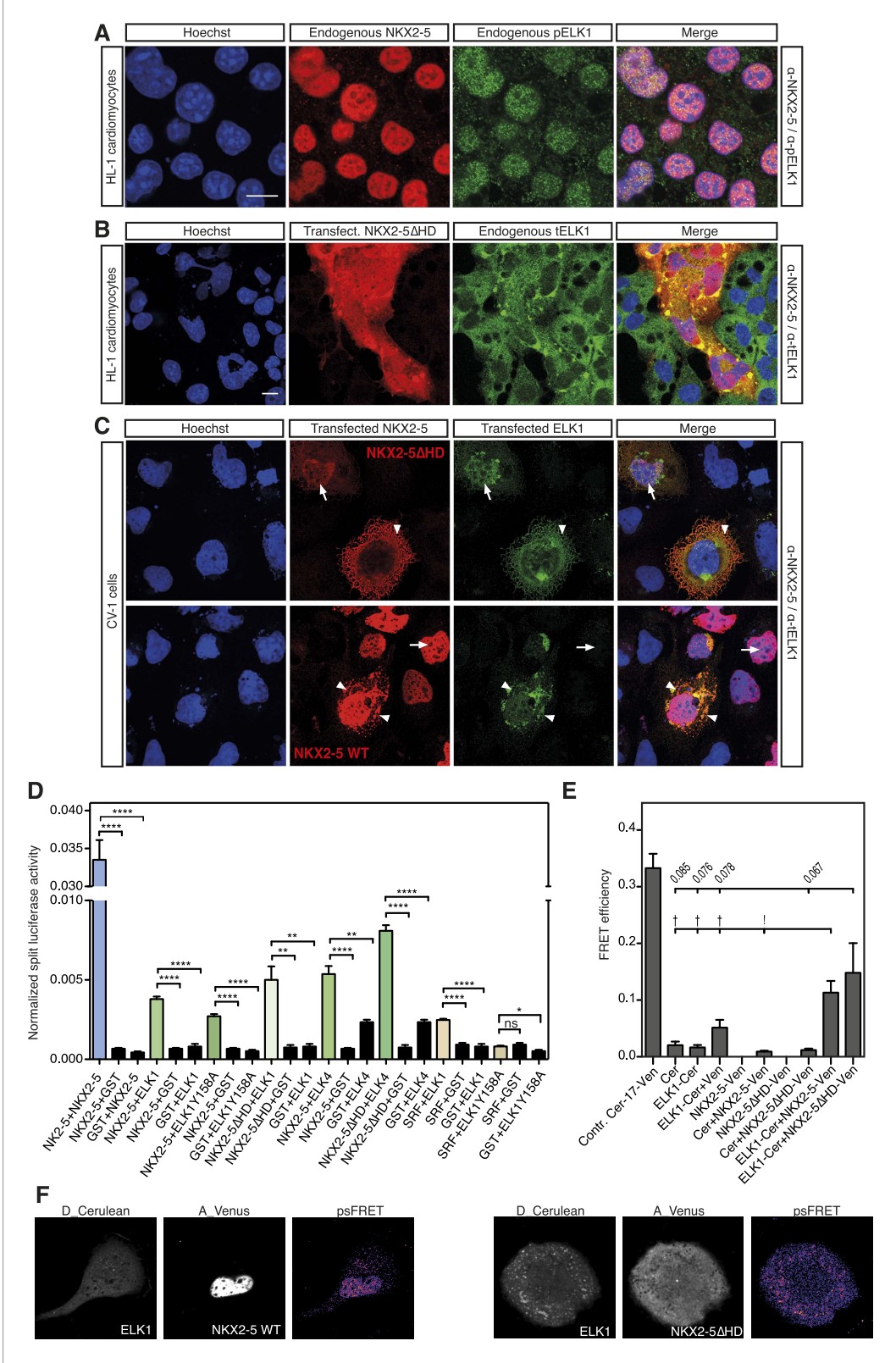

**Figure 5**. NKX2-5 WT and NKX2-5ΔHD interact with ELK1/4. (**A**) Nuclear expression of endogenous NKX2-5 and pELK1 in HL-1 cells. Scale bar represents 10 μm. (**B**) Co-localisation of transfected V5-tagged NKX2-5ΔHD and endogenous total ELK1 in HL-1 cells. Scale bar represents 10 μm. (**C**) Co-localisation of transfected HA-tagged ELK1

*Figure 5. continued on next page*

*Figure 5. Continued*

and V5-tagged NKX2-5ΔHD or NKX2-5 WT in CV-1 cells. Arrowheads and arrows show cells with cytoplasmic and nuclear NKX2-5 staining, respectively. (**D**) Interactions between NKX2-5, NKX2-5ΔHD, or SRF and ELK4, ELK1, or ELK1Y159A measured by the *R*luc-PCA in HEK 293T cells. (**E**) Interactions between ELK1 and NKX2-5 or NKX2-5ΔHD measured by psFRET in CV-1 cells. p-values: † <0.05; ! <0.001; values are given if p-value>0.05. (**F**) Representative psFRET images used in (**E**).

The following figure supplements are available for figure 5:

**Figure supplement 1**. HA-tagged ELK1 expression (green) induces nuclear translocation of V5-tagged NKX2-5ΔHD (red) in CV-1 cells.

**Figure supplement 2**. (**A**) Interactions between TBX5 and ELK1/ELK4 measured by the *R*luc-PCA in HEK 293T cells (** p < 0.01; *** p < 0.001; ****p < 0.0001).

peaks, 72 (14.2%) were bound by ELK1/4, representing more than half of the 26% of this C-set predicted by *Clover* to carry an ELK1/4 binding motif. By comparison, <1% of NKX2-5ΔHD C-set peaks overlapped with randomly generated peak sets. In the case of NKX2-5Y191C C-set peaks, 6.5% were bound by ELK1/4. These data provide strong support for our hypothesis that ELK1/4 directs NKX2-5 mutants to a subset of off-targets.

To explore the significance of ETS co-occupancy for selection of off-targets, we used DamID to determine the targets of an NKX2-5 mutant carrying tyrosine–alanine (Y-A) substitutions in the YRD because this mutant does not interact with ELK1/4. Comparisons between NKX2-5YRD$^{Y-A}$ and NKX2-5 WT targets revealed robust B and C-set (650 and 380 peaks, respectively; *Figure 2B* and *Supplementary file 1*). However, in the C-set, only 3.2% of peaks were occupied by ELKs. This lends further support for the notion that NKX2-5ΔHD binds a subset of its off-targets via cofactors interacting with the YRD. The proportionally large size of the NKX2-5YRD$^{Y-A}$ C-set also suggests that this mutation is comparable in severity to NKX2-5Y191C (*Figure 2B*), supporting the genetic evidence (*Elliott et al., 2006*).

## NKX2-5ΔHD is functional on target genes

Dysregulation of C-set genes may contribute to CHD. To test whether NKX2-5 mutants can influence the expression of their off-targets, we cloned the promoter regions of *Rad50* and *Snai2*, which bind to NKX2-5ΔHD, NKX2-5Y191C, ELK1, and ELK4, but not to NKX2-5 WT, into a luciferase reporter. In HEK 293 cells, both promoters were stimulated modestly by NKX2-5 WT but repressed by ELK1 or ELK4 (*Figure 6E*). NKX2-5ΔHD had no activity alone, but significantly enhanced repression by ELK1 or ELK4, an activity not displayed by NKX2-5 WT. NKX2-5Y191C had a similar activity, albeit weaker. These data show that NKX2-5ΔHD indirect binding to off-targets can modify gene expression, at least when over-expressed.

We next tested whether NKX2-5 mutants could alter off-target gene expression in the inducible ES cell lines described above. For off-targets from the NKX2-5WT:Y191C comparison, NKX2-5 WT modulated the expression of 6/22 C-set genes tested by > 1.5-fold (*Figure 6F*), although the effect on 2 of these (*Dock4* and *Sparc*) could be accounted for by NKX2-5 WT binding to a distinct DamID peak, and consistently, NKX2-5 WT and Y191C had opposite effects on these two genes. Thus, NKX2-5 WT influenced a minority of NKX2-5Y191C off-targets. NKX2-5Y191C repressed *Dock4* and activated 7/22 other C-set genes with 6 of these not regulated by NKX2-5 WT at the >1.5 fold significance threshold. Other genes showed a similar trend. Thus, NKX2-5Y191C could regulate ≥36% of its off-targets in this system.

## ELK1/4 are embedded within the cardiac GRN

As shown above, ETS motifs were enriched in NKX2-5ΔHD off-targets. They were also present although not significantly enriched in total NKX2-5 WT targets. Enrichment was also low in previously reported genome-wide NKX2-5 targets determined using ChIP-seq, but higher in targets of GATA4, SRF, and TBX5 (*He et al., 2011*). These and our DamID data suggest that ELK1/4 play a role in the normal cardiac GRN. To explore this further, we examined the overlap between ELK1/4 and NKX2-5

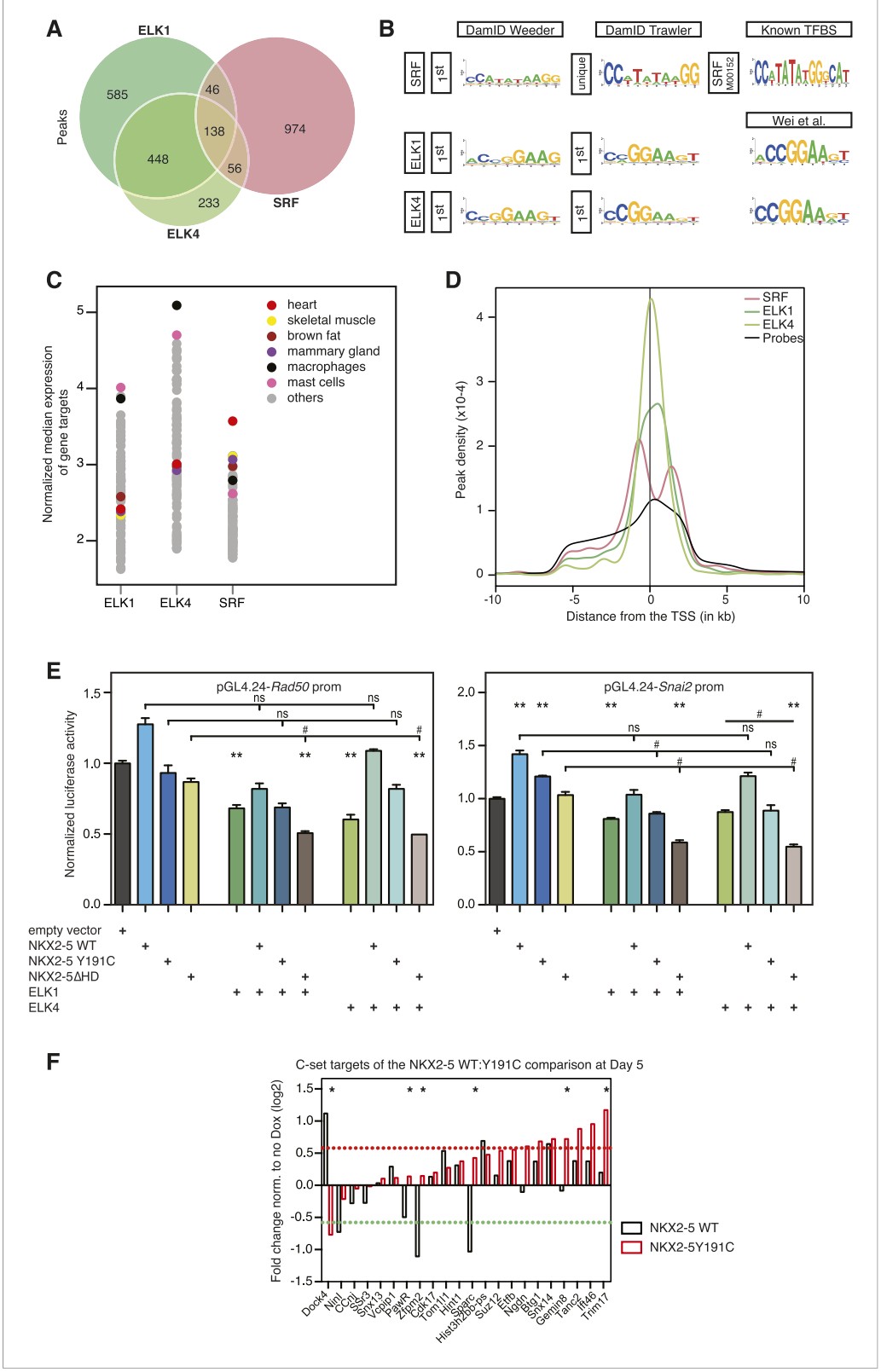

**Figure 6**. ELK1 and ELK4 co-occupy NKX2-5ΔHD off-targets in HL-1 cells. (**A**) Overlapping peaks between ELK1, ELK4, and SRF as shown by proportional Venn diagram. (**B**) Top binding motifs discovered de novo with *Weeder* or *Trawler* in ELK1, ELK4, and SRF peaks. The *TRANSFAC* SRF motif and the ELK1/4 motifs determined by (*Wei et al., 2010*) in vitro are shown on the right. (**C**) Normalised median expression of ELK1, ELK4, and SRF target genes in 91

*Figure 6. continued on next page*

*Figure 6. Continued*

murine cell types, including the heart in red (data collected from *BioGPS*). (**D**) Density of ELK1, ELK4, and SRF peaks and probes relative to the TSS. (**E**) Normalised *R*luc activity in HEK 293 cells. NKX2-5 WT and mutants were co-transfected with a pGL4.24 luciferase reporter under the control of the *Rad50-* or *Snai2*-promoters. p-values < 0.01 calculated relative to control are denoted by **. # shows significant difference (p < 0.01). (**F**) RT-PCR quantification of NKX2-5Y191C off-target genes in NKX2-5 WT or Y191C dox-inducible ES cell lines at day 5 (24 hr post-induction). Results were normalised to expression in uninduced cells. p-values were calculated using a t-test (* p < 0.05).

WT target genes. Of 1490 NKX2-5 WT target genes, 21% (p-value < 2.2e-16, Fisher exact test) were also bound by ELK1 or ELK4 (*Figure 7A*), and in 52% of these (including *Nkx2-5*, *Actc1*, *Id2*, and *Id3*), the NKX2-5 and ELK1/ELK4 peaks overlapped. Overlaps between SRF and ELK1 targets, and between SRF and NKX2-5 targets have been documented previously, although not in the same cell type (*Boros et al., 2009a*; *He et al., 2011*; *Schlesinger et al., 2011*). To complete our comparisons, we therefore determined SRF targets in HL-1 cells using DamID.

For SRF, we generated data for both N- and C-terminal Dam fusions. SRF peaks showed low false discovery rates (*Figure 1—figure supplement 4*) and significant overlaps between N- and C-terminal Dam fusions (p < 0.001), resulting in 1214 high-confidence peaks (*Figure 6A*; *Supplementary file 1*). These peaks could be assigned to 1314 unique target genes. De novo motif discovery identified the known SRF motif (CArG box) as the only over-represented motif (*Figure 6B*; *Supplementary file 1*) and present in 26% of targets, supporting the high specificity of DamID. We note that a previous ChIP study in HL-1 cells reported SRF targets with only a very small fraction containing the CArG box (*Schlesinger et al., 2011*). For SRF target genes, GO analysis showed high over-representation of cardiac (e.g., *heart development*) and generic (*cytoskeletal organisation* and *RNA processing*) terms (*Supplementary file 1*). SRF target median expression was highest in heart and skeletal muscle (*Figure 6C*), consistent with its known functions. Like NKX2-5 WT, SRF peaks displayed a bimodal distribution around the TSS (*Figure 6D*).

The overlap between ELK1/4 and SRF peaks was low (15 and 22% of ELK1 and ELK4 peaks, respectively; and 20% of SRF peaks; *Figure 6A*), consistent with previous ChIP studies in serum-starved HeLa cells suggesting that most ELK1 targets are not co-bound by SRF (*Boros et al., 2009a*). In HL-1 cells, targets co-occupied by ELK1/4 and SRF nonetheless included the *Fos* gene—the defining target of ternary SRF-ELK complexes (*Treisman, 1994*).

Targets bound uniquely by ELK1/4 or SRF showed over-representation of GO terms related to *catabolic* and *metabolic processes*, and as for total ELK1/4 targets, were only modestly expressed in heart (*Figure 7A*; *Supplementary file 1*). In contrast, targets bound by NKX2-5 only, NKX2-5, and SRF, or NKX2-5, SRF, and ELK1/4 were highly expressed in heart. NKX2-5 unique targets were enriched in GO terms *anion transport* and *regulation of cell motion*, while targets co-bound by NKX2-5 and SRF were enriched in GO terms related to *heart contraction*. Co-targets of all 3 factors (NKX2-5, SRF, ELK1/4) or of NKX2-5 and ELK1/4 were most over-represented in GO terms *heart development*, *blood vessel morphogenesis*, and *cytoskeleton organisation*. These data show that, overall, ELK1/4, SRF, and NKX2-5 regulate many generic cellular functions, while NKX2-5 and SRF also have more executive roles in regulating heart development and contraction. Furthermore, ELK1/4 appear to control very specific subsets of the cardiac GRN. By integrating the targets of all WT TFs profiled with those of other cardiac TFs previously published, we found that *Elk1/4* were embedded in the developmental cardiac GRN with many cross and feedback connections (*Figure 7B*), suggesting that ELK1/4 play a significant role in normal and disease cardiac GRN logic.

## Discussion

Here, we used DamID in HL-1 cells to probe the cardiac GRN by identifying target genes for kernel TFs NKX2-5 and SRF, and for the MAP kinase signalling-dependent cardiac transcriptional cofactors ELK1 and ELK4. To begin to understand the mechanism of CHD, we adopted the DamID method and identified targets of NKX2-5 mutants that mimic those found in CHD. These are the first functional genomics studies to probe CHD, providing new insights into the structure and function of NKX2-5, and network regulation in normal heart development and CHD.

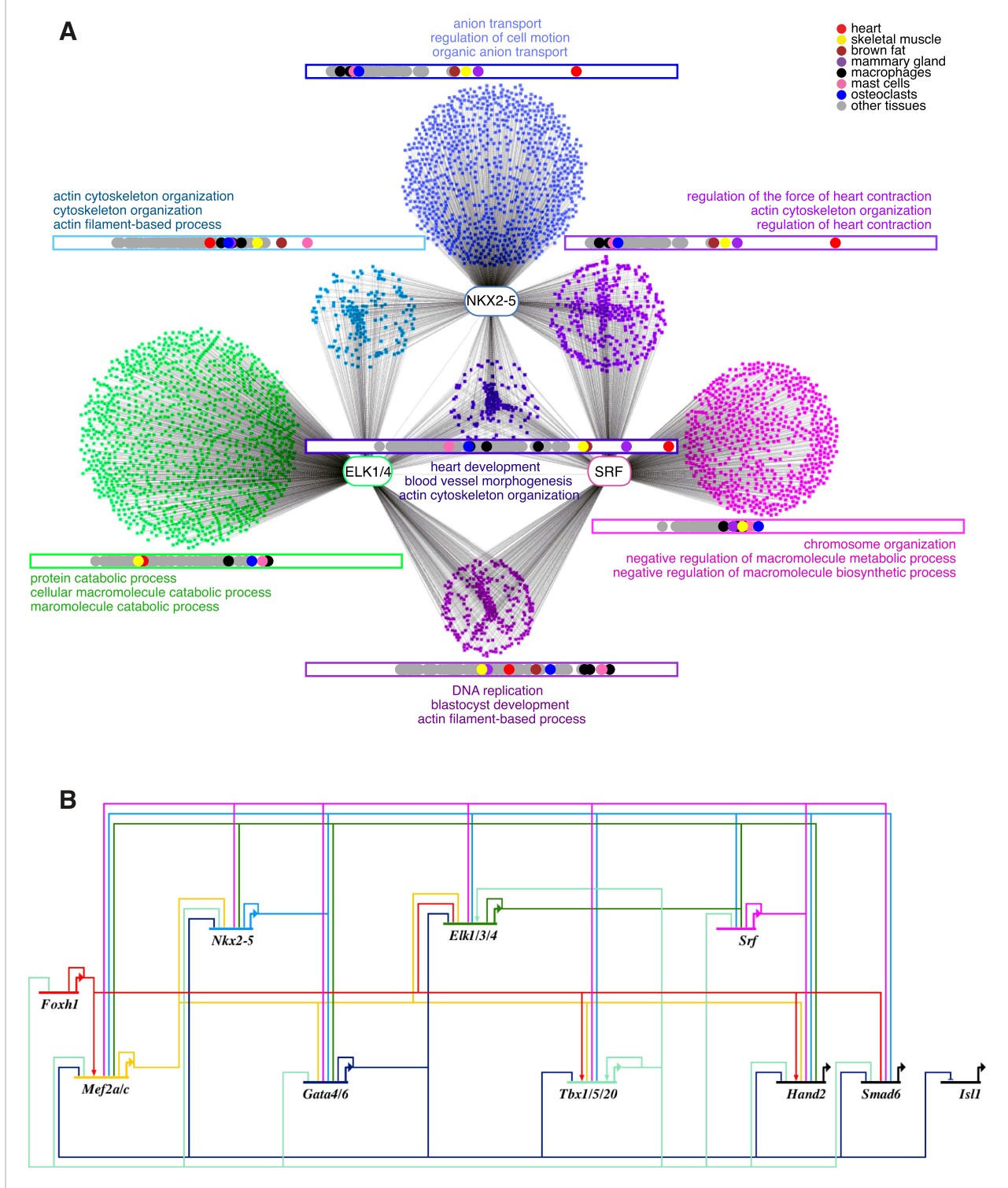

**Figure 7**. ELK1 and ELK4 are embedded in the cardiac gene regulatory network. (**A**) Overlapping target genes between NKX2-5, ELK1/ELK4, and SRF visualised using *Cytoscape* (spring-embed layout). Cluster size is proportional to gene numbers. For each cluster, rectangles indicate the normalised median expression of target genes in 91 murine cell types (*BioGPS*). Cell types are ordered by increasing expression values from left to right. Top 3 over-represented *DAVID* Gene Ontology (GO) annotations are indicated for each cluster. (**B**) Regulatory interactions between Elk1/3/4 and cardiac TFs from DamID experiments and published data sets (network constructed with *BioTapestry*).

Previous studies comparing DamID and ChIP experiments performed in *Drosophila* reported 'a high degree of overlap' (*Moorman et al., 2006*; *Negre et al., 2006*; *Tolhuis et al., 2006*; *van Bemmel et al., 2010*; *Yin et al., 2011*). We compared our NKX2-5 peak set with NKX2-5 ChIPseq peaks generated previously in HL-1 cells (*He et al., 2011*) and adult hearts (*van den Boogaard et al., 2012*), restricting our analyses to genomic regions covered by the promoter microarrays (*Figure 1—figure supplement 9*). We found that the overlap between all three data sets was low—specifically 18% and 6% DamID peak overlap, respectively. This low overlap highlights the problems of comparing data from different platforms and different laboratories, and warrants deeper analysis. We note that only DamID was performed with 3–4 replicates per experiment. Furthermore, de novo motif discovery identified the known high-affinity NKE exclusively within NKX2-5 DamID peaks. Both of the published studies identified a similar but variant NKE using ChIPseq. Cardiac- and muscle-related GO terms were more highly enriched in NKX2-5 target genes determined by DamID compared to published ChIPseq peaks from adult hearts included in *Figure 1—figure supplement 9* (*van den Boogaard et al., 2012*), and no such GO term were detected in data from HL-1 cells (*He et al., 2011*). These results provide strong validation for the DamID method.

Our study using DamID confirms for the first time at a genome-wide level in vivo, the prevailing view that severe mutations in cardiac kernel TF (such as NKX2-5ΔHD, Y191C, and YRD$^{Y–A}$) fail to bind the majority of WT targets (A sets) even in the presence of normal levels of NKX2-5 WT. Loss-of-binding of mutant proteins will be associated with alterations in the level of expression of many genes in the cardiac network through haploinsufficiency, which will be a major contributor to the structural and functional heart defects found in human CHD patients.

Surprisingly, however, NKX2-5 mutants still recognised a large number of genomic sites and some of these were normally bound by NKX2-5 WT (B-sets). B-sets included direct and indirect NKX2-5 WT target genes and were enriched in heart- and muscle-related GO terms. In the ES cell system, NKX2-5 mutant proteins retained WT-like functionality on some B-set genes. We propose that B-sets are recognised by mutants via dimerisation with cofactors or with NKX2-5 WT. The NKX2-5 HD is a recognised interface for NKX2-5 homo-dimerisation and interactions with cofactors, such as GATA4, HAND1, TBX5/20, and MEF2. NKX2-5ΔHD lacks the HD and embedded nuclear localisation domain and represents a severe mutation that has lost the ability to bind to the majority (81%) of normal NKX2-5 WT targets (A-set). However, even this severe mutant retained binding to a subset of direct and indirect NKX2-5 WT targets (B-set). NKX2-5ΔHD binding to B-set peaks could *only* occur through heterodimerisation with NKX2-5 WT or cofactors. We have shown using a number of approaches that NKX2-5ΔHD retained its ability to interact with NKX2-5 WT, GATA4, and TBX5/20, and we identified the YRD located within the C-terminus of NKX2-5 as a novel interface that collaborates with the HD to support these interactions. Our characterisation of the YRD as a novel homophilic and heterophilic interaction domain provides the molecular mechanism that explains dimerisation of NKX2-5ΔHD to NKX2-5 WT or its cofactors in the absence of the HD. We predict that the HD and YRD act supportively and synergistically in the assembly of macromolecular TF complexes. The YRD is an ancient domain that has coevolved with the NK2-class HD and is essential for NKX2-5 function in early mouse embryogenesis (*Elliott et al., 2006*). In humans, half of NKX2-5 mutations associated with CHD are outside of the HD, and three frame-shift mutations located within the YRD have been identified (*Benson et al., 1999*; *Gutierrez-Roelens et al., 2002*; *Ikeda et al., 2002*).

Both NKX2-5 mutants studied also recognised a large number of targets uniquely (C-set), which we term off-targets. The relative size of the C-set within the total mutant target set (C-set/B-set ratio) correlates with the location and predicted severity of NKX2-5 mutations (ΔHD>Y191C≥ YRD$^{Y–A}$), and thus B-set and C-set size and content may prove to be valuable signatures for understanding the relationship between genotype, phenotype, and clinical outcomes in CHD. NKX2-5ΔHD off-targets were associated with GO terms *chromatin organisation* and *cell cycle*, and peaks were enriched in the binding sites for ETS family and many other broadly expressed TFs, attesting to an underlying logic in their selection. A high percentage of NKX2-5 mutant off-targets (14.2% for NKX2-5ΔHD and 6.5% for NKX2-5Y191C) were occupied by ELK1/4 in HL-1 cells, compared to <1% when randomly generated peaks were overlapped. Furthermore, only 3% of C-set peaks of the NKX2-5YRD$^{Y–A}$ mutant were bound by ELK1/4. Although ETS factors have previously been predicted to be involved in heart development based on motif enrichment in predicted cardiac enhancers in fly and humans (*Pham et al., 2007*; *Narlikar et al., 2010*; *Jin et al., 2013*), this is the first evidence showing a direct interaction with cardiac TFs. Studies of the *Ciona intestinalis* cardiac GRN demonstrated that ETS

family gene *Ets1/2* is activated by FGF/MAP Kinase signalling, which specifies the identity of founder cells of the cardiac lineage (*Davidson et al., 2006*). Interestingly, most overrepresented in candidate Ets1/2 target genesis was the motif 5′ATTA3′, (*Davidson et al., 2006*; *Woznica et al., 2012*), which is similar to the HOX-like motif overrepresented in NKX2-5Y191C off-target peaks in HL-1 cells, potentially indicating a conserved FGF/MAP Kinase-activated synergy between ETS and HD factors at the earliest stages of cardiac lineage specification.

ELK1 and ELK4 interacted with NKX2-5 WT and NKX2-5ΔHD through the YRD, and this interaction was sufficiently strong for pELK1 to transport NKX2-5ΔHD into the nucleus when co-expressed and for unphosphorylated ELK1 to tether NKX2-5 WT in cytoplasmic inclusions. This interaction provides a mechanism for how severe NKX2-5 mutants lacking a functional HD can be directed to a sub-set of off-targets via interaction with broadly expressed TFs. Severe NKX2-5 mutants may be drawn to off-targets due to the breakdown of compelling cardiac specificity on true targets.

We have also shown that ETS factors ELK1 and ELK4 are normal cofactors of NKX2-5, embedded in the cardiac GRN kernel. The cardiac GRN kernel is traditionally conceptualised as a set of cardiac-restricted TFs that are critical for defining organ territories and imposing organ specificity to GRN logic. However, the cardiac kernel is likely to have evolved in the context of a host of ancient TFs controlling ubiquitous cellular processes such as metabolism, cell cycle, chromatin dynamics, and cytoskeleton. Furthermore, it is well known that TFs are amongst the targets of the complex paracrine signalling pathways that underpin pattern formation, tissue specification, and differentiation in metazoans. Our genome-wide studies suggest that ELK1/4 interact with, regulate, and are regulated by, cardiac TFs, and are therefore recursively wired into the cardiac kernel at a high level. Other studies show that organ-restricted NK-2 class HD TFs collaborate with ETS factors on specific target genes (*Chen and Bieberich, 2005*; *Lin et al., 2006*) or provide signal gating for decisions affecting lineage fate within developmental fields (*Halfon et al., 2000*; *Hollenhorst et al., 2011*). ELK1/4 bind and potentially regulate a large number of genes in HL-1 cells involved in ubiquitous cellular processes, most prominently metabolism. However, smaller gene sets, most significantly those encoding heart developmental and cytoskeletal genes, were co-bound by NKX2-5 and ELKs, SRF and ELKs, or ELK1/4, NKX2-5, and SRF. One interpretation is that subsets of ELK1/4 targets, potentially those at critical hubs, have been drawn into the cardiac GRN to exert strategic (signal-gated) and fine network control over its outputs. Considering the implications of this and related studies more broadly, network links between cardiac and canonical signal-gated TFs are likely to be much greater than currently appreciated and generate massive potential for regulatory coding that needs to be tested in future studies.

Our results may have implications for the mechanism of CHD. While we acknowledge that our findings are derived from studies in a cell culture system, they compellingly suggest that NKX2-5 mutants, even those lacking DNA-binding ability, can bind to a subset of normal targets and off-targets, where they could affect transcription either positively or through dominant-negative action. Importantly, both NKX2-5ΔHD and NKX2-5Y191C retained some functionality on select B- and C-set targets in vitro, an activity driven by heterodimerisation with cofactors. Our results confirm the inability of severe mutants to bind most normal targets, and therefore that haploinsufficiency is likely the dominant component of CHD mechanism. However, we predict that allele-specific dominant-negative and gain-of-function effects arising from dysregulation of the hundreds of off-targets will further destabilise the cardiac GRN and could contribute to disease. We have not yet analysed the expression of off-targets in mouse models or human CHD samples. Defining the targets and off-targets of a range of human CHD mutations and analysing their expressions in appropriate animal and human models will be needed to rigorously test the hypothesis.

## Materials and methods

### Plasmids and cell culture
#### Plasmids/cloning
Sequences coding for murine TFs used in this study were amplified from HL-1 cell cDNA and cloned into DamID vectors generated previously (*Vogel et al., 2006*). The mouse NKX2-5Y191C mutant was generated by site-directed mutagenesis of the *Nkx2-5* cDNA using the following primer and its reverse complement: TCCAGAACCGTCGCTGCAAGTGCAAGCGACAG. The NKX2-5ΔHD (NKX2-5

V142_Q199del) was constructed by replacing the HD with 8 glycine residues. Vectors used for lentivirus production were obtained from Prof D Trono/Addgene: pMLDg/pRRE (12,251), pRSV-Rev (12,253), and pMD2.G (12,259) (*Dull et al., 1998*).

## Cell lines

The HL-1 cell line was donated by Prof W C Claycomb (Department of Biochemistry and Molecular Biology, Louisiana State University, New Orleans, LA, USA) (*Claycomb et al., 1998*). Human embryonic Kidney 293 FT and Ecr 293 cells were from Invitrogen.

For doxycycline (dox)-inducible mouse ES cell lines, *Nkx2-5* WT, *Nkx2-5$^{\Delta HD}$*, and *Nkx2-5$^{Y191}$*, open reading frames were amplified by PCR from the plasmids described in section 'Plasmids/cloning', sequence-verified and cloned in place of Mesp13XFlag in the p2LoxMesp1-3XFlag-IRES-EGFP vector (*Bondue et al., 2008*). These constructs were electroporated in A2Lox cells, and stable cell lines were selected (*Bondue et al., 2008*).

ES cells were cultured on irradiated mouse embryonic fibroblasts (*Bondue et al., 2008, 2011*). Before differentiation, one ES cell passage was performed on gelatine-coated plates to eliminate remaining fibroblasts. ES cell differentiation was performed in re-aggregated EBs in serum-free conditions (*Kattman et al., 2011b*). Briefly, to allow their re-aggregation, ES cells were dissociated with trypsin and grown in Corning ultra-low adhesion plates at a density of 75,000 cells/ml for 48 hr without growth factors. After 2 days, the medium was changed and growth factors added (12.5 ng/ml BMP4, 6 ng/ml Activin, and 5 ng/ml VEGF) for 48 hr. At day 4, EBs were collected and replated on gelatine-coated plates in a StemPro34-based medium (Life Technologies, Carlsbad, USA), supplemented with 100 U/ml Penicillin (Life Technologies), 100 µg/ml Streptomycin (Gibco), 2 mM L-Glutamine, 50 µg/ml ascorbic acid (Sigma-Aldrich, St Louis, USA), 10 ng/ml bFGF, 25 ng/ml FGF10, 5 ng/ml VEGF, and 5 µM IWR1 (*Willems et al., 2011*; *Kattman et al., 2011a*). All growth factors were purchased from R&D, except for IWR1 obtained from Sigma. After replating, medium was replaced at day 6 and 8 of differentiation. When indicated, dox was added at day 4 at a final concentration of 1,000 ng/ml (*Bondue et al., 2008*).

## Immunofluorescence and imaging

Immunofluorescence on fixed cells was performed as described by (*Chapman et al., 2011*). Antibodies used are listed in *Supplementary file 2A*.

## Transactivation experiments in mammalian cells

Transactivation experiments were performed in HEK 293 FT cells (*Costa et al., 2011*). We used the following primers for *cis*-element PCR-amplification from mouse HL-1 cell DNA: ACGCTCCTGACTT GACTGTTTT and GCAGCGGCCGACTCTTATAG for the *Id3* promoter; CCCGAGTCCCTTGGCTAACT and AACCCCAGCCCTTCCTACTAAC for the *Id3* downstream region (DR); ACCCTGCAGGACAT GAACTCA and TGGGCTAGGAGGCTCTGTAATA for *Rad50*; and GCGCTACAAAGGGAGGAAGTC and CATGCAGGAGACTCCATAGGAG for *Snai2*. All cis-elements were cloned into the pGL4.24 Luciferase reporter vector (Promega, Madison, USA).

## Gene expression by real-time-PCR

For gene expression analysis in mouse embryos, total RNA was extracted from dissected hearts from embryos aged between 12 and 20 somite pairs (E8.5–E9.0) using the Stratagene Absolutely Nanoprep Kit (Agilent Technologies, Santa Clara, USA). Six hearts for each genotype wild type *Nkx2-5$^{+/+}$*, *Nkx2-5$^{LacZ/+}$* heterozygous, and *Nkx2-5$^{LacZ/LacZ}$* (*Elliott et al., 2006*) were pooled for reverse transcription into cDNA using Superscript III (Life Technologies) with poly-(dT). Quantitative PCR was performed with SYBR green (Roche, Basel, Switzerland) and analysed on the Roche LightCycler480. Levels of gene expression were normalised to *Hprt*. Primers are listed in *Supplementary file 2B*.

For gene expression analysis in mouse ES cells, RNA extraction, DNase treatment, and RT-PCR were performed as previously described (*Bondue et al., 2008, 2011*). qPCR was performed using Brilliant II Fast SYBR qPCR Master Mix on a Mx3005P Real-Time PCR system (Agilent). All primers were designed using Lasergene 7.2 software (DNAStar Inc, Madison, USA) and are listed in *Supplementary file 2B*. Gene expression was normalised to *Tbp* and *Actb*.

## Electrophoretic mobility shift assay

Electrophoretic mobility shift assays (EMSAs) were performed according to (*Watanabe et al., 2012*). Protein production was done according to manufacturer's recommendations using the Promega TnT

Quick Coupled Transcription/Translation System. Binding reactions were performed in 75 mM NaCl, 10 mM Tris-Cl pH7.5, 1 mM EDTA, 1 mM DTT, 6% Glycerol, and 1 mM MgCl2 on ice for 60 min. We used the following oligonucleotides and their reverse complements: GGGACCTTTGAAGTGGGGGCCTC for the Nppa-NKE; GGGCTGCTTCTGGCAGAATGGAG for NF-1 #I; and GGGCCTCTTCTGGCAGGAGG GAG for NF-1 #II.

## Animal experimentation

Animal experimentation was performed with approval of the Garvan Institute/St Vincent's Hospital Animal Ethics Committee (Project numbers 10/19 and 10/01).

## ChIP experiments

ChIP experiments were performed in HL-1 cells following the Abcam Cross-link Chromatin (X-ChIP) protocol. Proteins and DNA were cross-linked with 0.75% formaldehyde at room temperature for 10 min. Immunoprecipitation of chromatin complexes was done using Santa Cruz antibodies cat no sc365207 for NKX2-5 and sc2025 for IgG control. Target gene levels were quantitatively measured by PCR. Primers were designed to validate DamID peaks and are provided in *Supplementary file 2B*.

## DamID experiments

DamID experiments were performed after modification of published protocols (*Vogel et al., 2007*). In brief, HL-1 cells at confluency were transduced with lentiviral vectors allowing undetectable expression of TF/Dam fusion proteins in the absence of induction. After 40 hr, genomic DNA was extracted using a Gentra PureGene Cell kit (QIAGEN, Venlo, Netherlands), digested by DpnI at 37°C for 6 hr, and amplified by ligation-mediated PCR. PCR products were further fragmented with DNaseI at 24°C for 1 min, and labelled and hybridised to Affymetrix mouse 1.0R promoter microarrays according to manufacturers' instructions. Three independent DamID experiments were performed utilising biological triplicates in first two studies and quadruplicates in the third study (*Supplementary file 2C*). Peaks can be directly visualised in the UCSC Genome Browser following this http://genome.ucsc.edu/cgi-bin/hgTracks?db=mm9&type=bed&hgt.customText=ftp://ftp.ncbi.nlm.nih. gov/geo/series/GSE44nnn/GSE44902/suppl/GSE44902_DamID.bed.gz. Microarray data were deposited in NCBI's Gene Expression Omnibus (GSE44902).

## Bioinformatics analysis

### Array processing and binding loci (peaks) detection

Quality control was first performed using the 'affy' and 'affyPLM' R packages to assess the distribution of probe-levels effects via the RLE and PLM methods (*Gautier et al., 2004*). RB_NS12.CEL was found to have elevated NUSE (median away from 1) and RLE (median away from 0). RB_AW10.CEL was found to have elevated RLE (median away from 0) and an artifact identified by pseudoimage decomposition of the microarray. Log2 transformed microarray sample probe set intensities normalised via Robust Multichip Average and followed by complete linkage clustering (Pearson correlation) also identified RB_NS12.CEL as an outlier. RB_AW10.CEL and RB_NS12.CEL were removed from subsequent analyses. Probe remapping to NCBI Build 37 (mm9) was performed using the Starr R package (*Zacher et al., 2010*). Inputs to Starr for remapping were the probe mappings for the Affymetrix Promoter 1.0R Array and all chromosomes from the NCBI Build 37 (mm9) in Fasta format downloaded from the UCSC ftp server. Array normalisation and peak identification were performed using the CisGenome software package version 2.0 (*Ji et al., 2008*). Arrays were normalised using the quantile normalisation method implemented in CisGenome. The TileMapv2 algorithm was then used to detect regions that bound at least 8 probes with a moving average statistic ≥3.5, all other parameters remained default (*Ji and Wong, 2005*).

Microarrays (.CEL files) and statistical output from CisGenome (.COD files) were deposited in NCBI's Gene Expression Omnibus, accession number GSE44902 (*Edgar et al., 2002*). A bed file of all peak coordinates called using CisGenome was compiled and customised for input to UCSC genome browser (mm9) and available for download from NCBI GEO accession number GSE44902 (*Bouveret et al., 2015*). Alternatively, these data can be automatically displayed by clicking the link above.

## Gene assignment and binding site locations

Peaks were assigned to genes using GREAT (*McLean et al., 2010*) with the following parameters: Species Assembly: Mouse mm9. Background regions: Custom Perl scripts were written to fetch non-coding regions from 6.5 kb upstream and 2.5 kb downstream of all EnsEMBL genes (API version 66, mm9) (*Flicek et al., 2012*) that are covered by probes on the Affymetrix promoter array. Association rule settings: Basal plus extension, 6.5 kb upstream, 2.5 kb downstream up to 100 kb. Custom Perl scripts were written to locate the distance of the median position of each peak from the closest annotated transcription start site (annotations also obtained through the EnsEMBL API). Position frequencies were plotted using R (http://www.r-project.org/). Peak overlaps were calculated using the BEDTools package using a default minimum of 1 bp overlap (*Quinlan and Hall, 2010*). For comparison with data from (*He et al., 2011*) and (*van den Boogaard et al., 2012*), we obtained NKX2-5 peaks determined by the authors. Gene overlaps were visualised using BioVenn (*Hulsen et al., 2008*).

## GO analysis

Functional significance of peaks (TF-binding regions) was assessed using GREAT (*McLean et al., 2010*). Significantly enriched GO terms were defined as those with p-value ≤ 0.05 and having at least 4 target genes associated with that term. Functional significance of genes (determined from overlapping annotations) was performed using DAVID version 6.7 (current release January 2010) (*Huang da et al., 2007*; *Huang da et al., 2009*).Background used to identify enriched GO terms with GREAT was all genes that could be assigned from the probes on the Affymetrix microarrays. The whole genome was used as background with DAVID.

## DNA motif discovery

De novo motif analysis of TF-binding sites was performed using the stand-alone version of *Weeder* 1.4.2 (*Pavesi and Pesole, 2006*) and Trawler_standalone (*Haudry et al., 2010*). Parameters for running *Weeder* were: search mode=small-medium. Parameters for running Trawler_standalone were: motif_number=200-1000; mlength=6-12; wildcard=23; strand=double; overlap=70-90; occurrence=10; other options remained default. Repeat-masked background sequences from 6500 bp upstream to 3500 bp downstream of each annotated transcription start site were retrieved from EnsEMBL using custom Perl scripts (API version 66, mm9) (*Flicek et al., 2012*).

Clover (*Frith et al., 2004*) was used to find enrichment of known TF-binding motifs in DamID peaks. The background sequences used were the regions from 6.5 kb upstream to 3.5 kb downstream of all genes tiled on the Affymetrix Mouse Promoter Tiling Array. Motifs with p-values<0.05 were deemed significantly over-or under-represented.

## Tissue expression analysis

A pre-processed series matrix file containing the expression profiles of 91 mouse cell and tissue types was downloaded from the NCBI Gene Expression Omnibus (GSE10246). Replicates were transformed to log2 normalised mean expression (NME), and gene symbols were assigned using the mouse4302.db library in R. Using the background gene assignments from section 'GO Analysis', 1000 genes were drawn at random, and the median of each tissue/cell expression profile for these genes was calculated from the NME of all samples. This was repeated 100 times and the average taken to form a static random background expression profile. For any given set of input genes, a tissue/cell type expression profile was formed by the ratio of the randomly generated median expression profiles to the median tissue/cell type expression profile of the input group of genes.

## Gene network reconstruction

*BioTapestry* (*Longabaugh et al., 2009*) was used to represent the protein-DNA interactions between cardiac TFs identified in this study and elsewhere (*von Both et al., 2004*; *Davidson and Erwin, 2006*; *Mori et al., 2006*; *Rojas et al., 2008*; *Silvestri et al., 2008*; *Boros et al., 2009b*; *Holler et al., 2010*; *van Bueren et al., 2010*; *He et al., 2011*; *Schlesinger et al., 2011*; *Shen et al., 2011*). Gene targets of NKX2-5, ELK1/4, and SRF obtained from the DamID analysis were visualised with *Cytoscape* using the spring embedded layout with default parameters (*Killcoyne et al., 2009*). Each TF surveyed by DamID was reanalysed with the spring embedded layout resulting in a surface that scaled proportionally to the number of genes connected to the TF surveyed.

## paFCS

HL-1 cells were seeded onto Lab-Tek chambers (Thermo Scientific, Waltham, USA) with 5 mg/ml fibronectin and 20 mg/ml gelatine and transfected with indicated paGFP fusion proteins and histone 2B (H2B)-RFP (to identify nuclei) using Lipofectamine 2000 (Invitrogen), as per manufacturer's instructions. Fluorescence correlation spectroscopy (FCS) measurements were performed 48 hr post-transfection, following photoactivation of the paGFP fusion proteins with 405 nm light, as previously described (*Kaur et al., 2013*) using the Zeiss LSM780 laser scanning confocal microscope, with the avalanche photodiodes of the Confocor 3 module (Zeiss, Jena). The autocorrelation function (ACF) G(t) for the fluorescent intensity over the 20 s measurement was calculated using the ZEN software FCS module (Zeiss, Jena, Germany). Mean ACFs (of 7 repetitions) were fitted using the ZEN Software (Zeiss) using a model that comprised two 3D diffusion terms, one free and one anomalous, and another term to account for triplet state photophysics (see [*Kaur et al., 2013*] for further details).

## Protein–protein interactions

### Split-luciferase protein fragment complementation assay

The *Renilla* luciferase protein fragment complementation assay (*R*luc-PCA) was performed based on a modified version of the technique described by *Stefan et al. (2007)*. Protein-coding sequences were cloned into a pcDNA3 expression vector backbone, which was modified to contain either *Renilla* luciferase fragment 1 (Rluc8.1, amino acids 1–110) or *Renilla* luciferase fragment 2 (Rluc8.2 amino acids 111–312), each followed by a 2×(GGGS) linker, upstream of a Gateway cloning cassette allowing insertion of the gene of interest. The resulting fusion is under the control of a truncated form of the CMV promoter (CMVd1), allowing for reduced levels of protein expression (Promega). Transfection efficiency was normalised by a third vector encoding LacZ, downstream of a CMV promoter (CMV-LacZ). The day before transfection, 20,000 HEK 293T cells were plated in clear-bottomed, white-walled 96-well plates in 100 ml DMEM/10% FCS growth media. A master mix for each quadruplicate was prepared, containing 16 µl Optimem (Life Technologies), 40 ng CMV-LacZ, 180 ng R.luc8.1

Fusion, 180 ng R.luc8.2-Fusion, and 0.8 µl FuGeneHD (Roche). The mix was incubated for 10 min, and equal amounts of mix added to each well of the quadruplicate. After 24 hr, cells were washed in PBS and 100 µl of PBS added to each well. A 10 mM solution of Benzyl Coelenterazine in PBS (Nanolight Technologies, Pinetop, USA) was prepared from a 10 mM stock solution in methanol, and 100 µl was injected into each well by luminometer (FluoStar Optima, BMG Labtech). Readings were taken for 5 s, 2 s after injection, and again for 5 s 15 min after initial injection of substrate. Following readings, cells were washed with PBS and ß-galactosidase detected using the Galacto-Star System (Life Technologies). *Renilla* luciferase readings were then normalised to levels of ß-galactosidase and the four replicates averaged. As Glutathione transferase dimerises, fusions to Rluc8.1 and Rluc8.2 were used as a positive control and were used separately with other proteins of interest as negative controls.

### FRET

CV-1 cells were seeded onto coverslips, and upon reaching ~70% confluency were transfected with the appropriate plasmids encoding Cerulean- and Venus-TF fusions, following the protocol supplied with Lipofectamine LTX reagent (Life Technologies). Briefly, for each well of a 6-well plate containing 2 ml of DMEM (minus FBS), 500 µl of Optimem was added to each plasmid mixture and 2.5 µl of Plus reagent added. This mixture was left to incubate at room temperature for 5 min, before 6.25 µl Lipofectamine LTX was added to each DNA/Optimem/Plus reagent mixture.

Following 30 min incubation at room temperature, 500 µl of the resulting DNA-lipid complexes were added drop wise to the appropriate well. Complexes were left on cells for 4–5 hr, before the serum-free media was replaced with DMEM containing 10% FBS and incubated 16–24 hr. Prior to live imaging, media were changed to phenol-red free DMEM, and the coverslip was mounted in a POCmini-2 cultivation system (Zeiss) in DMEMgfp medium (Evrogen, Moscow, Russia) containing 10% FCS on a LSM 7Duo confocal microscope equipped with a 32-channel spectral detector. 12-bit spectral images were acquired using a 63× Plan-Apochromat 1.40 oil objective at a spectral resolution of 10 nm, first with 405 nm laser excitation and then with a 514 nm laser. Donor, raw FRET, and acceptor images were obtained by linear-unmixing spectral data using newly acquired reference spectra in the Zen 2009 software (Zeiss). Images with average pixel values above 300 and below 2900 were processed through the psFRET algorithm to remove donor spectral bleed-through from the raw FRET signal (*Chen et al., 2007*).

## Yeast two-hybrid assay

Protein-coding sequences were cloned into pGADT7 AD or pGBKT7 DBD expression vector backbones (Clontech, Mountain View, USA), which were modified to contain a Gateway cloning cassette. pGADT7-AD and pGBKT7-DBD fusions were co-transformed into chemical competent *S. cerevisiae* strain AH109 (Clontech). Double transformants were selected for growth on -Leu/-Trp selection plates, before being selected for interaction on -Ade/-His/-Leu/-Trp selection plates.

## Acknowledgements

We thank MW Costa, A Ayer, J Stoeckli, D Trono (Addgene), J Lopez, and B van Steensel for vectors and technical assistance; WC Claycomb for HL-1 cells; G Arndt (Johnson&Johnson) for HEK Ecr-293 cells (Invitrogen); and H Speirs (Ramaciotti Centre for Gene Function Analysis) for microarray hybridisation. Work was supported by funds from the Office of Health and Medical Research, NSW State Government; by a donation from H Smith and grants from the National Health and Medical Research Council, Australia (NHMRC; 573732, 573703, 1052171, 514900, 1042002, 1061539), Australian Research Council (ARC; DP0988507, DP120104594, DE120100794), Atlantic Philanthropies (19131), British Heart Foundation (CH/09/003) and Wellcome Trust (083228). RPH held an NHMRC Australia Fellowship (573705); RB held Swiss National Science Foundation (PBEZB-111553), UNSW Vice Chancellor's, and ARC Postdoctoral Fellowships; MR held fellowships from EMBO (1133–2009), Human Frontier Science Program (HFSP; LT000245/2010-L), and NHMRC/National Heart Foundation (1049980); NS held an HFSP Fellowship (LT00044/2007-L).

## Additional information

### Funding

| Funder | Grant reference | Author |
|---|---|---|
| National Health and Medical Research Council (NHMRC) | 1061539 | Richard P Harvey |
| Atlantic Philanthropies | 19131 | Richard P Harvey |
| British Heart Foundation | CH/09/003 | Shoumo Bhattacharya |
| Wellcome Trust | 083228 | Shoumo Bhattacharya |
| Schweizerische Nationalfonds zur Förderung der Wissenschaftlichen Forschung | PBEZB-111553 | Romaric Bouveret |
| Australian Research Council (ARC) | DP0988507 | Richard P Harvey |
| European Molecular Biology Organization (EMBO) | 1133-2009 | Mirana Ramialison |
| Human Frontier Science Program (HFSP) | LT00044/2007-L | Nicole Schonrock |
| National Heart Foundation of Australia | 1049980 | Mirana Ramialison |
| National Health and Medical Research Council (NHMRC) | 573732 | Richard P Harvey |
| National Health and Medical Research Council (NHMRC) | 573703 | Richard P Harvey |
| National Health and Medical Research Council (NHMRC) | 1052171 | Nicolas Plachta |
| National Health and Medical Research Council (NHMRC) | 514900 | Sally L Dunwoodie |
| National Health and Medical Research Council (NHMRC) | 1042002 | Sally L Dunwoodie |
| National Health and Medical Research Council (NHMRC) | 573705 | Richard P Harvey |
| Australian Research Council (ARC) | DP120104594 | Nicolas Plachta |
| Australian Research Council (ARC) | DE120100794 | Nicolas Plachta |

| Funder | Grant reference | Author |
| --- | --- | --- |
| Human Frontier Science Program (HFSP) | LT000245/2010-L | Mirana Ramialison |

The funders had no role in study design, data collection and interpretation, or the decision to submit the work for publication.

## Author contributions
RB, Conception and design, Acquisition of data, Analysis and interpretation of data, Drafting or revising the article, Contributed unpublished essential data or reagents; AJW, NS, MR, Conception and design, Acquisition of data, Analysis and interpretation of data, Drafting or revising the article; TD, DJ, NP, Acquisition of data, Analysis and interpretation of data; AB, GK, Acquisition of data, Analysis and interpretation of data, Drafting or revising the article; SM, HF, Collected data, Acquisition of data; C-C, SB, Conception and design, Contributed unpublished essential data or reagents; MAW, Critically reviewed the study proposal, Contributed unpublished essential data or reagents; SLD, Acquisition of data, Drafting or revising the article; GC, Conception and design, Acquisition of data, Analysis and interpretation of data; CB, Conception and design, Analysis and interpretation of data; RPH, Conception and design, Analysis and interpretation of data, Drafting or revising the article

## Author ORCIDs
Merridee A Wouters, http://orcid.org/0000-0002-2121-912X

## Ethics
Animal experimentation: Animal experimentation was performed with approval of the Garvan Institute/St Vincent's Hospital Animal Ethics Committee (Project numbers 10/19 and 10/01).

## Additional files

### Supplementary files
• Supplementary file 1. Targets of NKX2-5, ELK1, ELK4, and SRF identified by DamID in murine HL-1 cardiomyocytes. This file contains genomic coordinates of peaks bound by NKX2-5, NKX2-5Y191C, NKX2-5ΔHD, ELK1, ELK4, and SRF and their respective target genes and GO over-representations determined using *GREAT* (*McLean et al., 2010*). This file also contains results of our de novo motif discovery analysis.

• Supplementary file 2. Supplementary tables. List of antibodies (**A**) and primers used for qPCR and ChIP (**B**). (**C**) Peak calling from three independent Dam experiments.

### Major datasets
The following datasets were generated:

| Author(s) | Year | Dataset title | Dataset ID and/or URL | Database, license, and accessibility information |
| --- | --- | --- | --- | --- |
| Bouveret R, Waardenberg AJ, Schonrock N, Ramialison M, Doan T, de Jong D, Bondue A, Kaur G, Mohamed S, Fonoudi H, Chen C, Wouters M, Bhattacharya S, Plachta N, Dunwoodie SL, Chapman G, Blanpain C, Harvey RP | 2015 | NKX2-5 mutations causative for congenital heart disease retain functionality and are directed to hundreds of targets | https://www.ncbi.nlm.nih.gov/geo/query/acc.cgi?acc=GSE44902 | Publicly available at the NCBI Gene Expression Omnibus (Accession number: GSE44902). |

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
