## [Decision Letter]

Thank you for sending your work entitled “NKX2-5 mutations causative for congenital heart disease retain functionality and are directed to hundreds of targets” for consideration at *eLife*. Your article has been favorably evaluated by Janet Rossant (Senior editor) and four reviewers, one of whom is a member of our Board of Reviewing Editors.

This paper makes an important contribution to the field of cardiogenesis and to understanding NKX2-5 driven mechanisms underlying congenital heart disease. It leads to a shift in the conceptual framework, from the conventional view that mutations in NKX2-5 result in a failure of the transcription factor to correctly activate its downstream targets to the idea that the mutant factor participates in promiscuous interactions that lead to inappropriate activation of genes that are not normally NKX2-5 targets. This is a very data rich study that represents a technical tour de force. The results are complex and the manuscript densely written. In order to make it more accessible, some simplifications and clarifications are required in the text. Further experimentation is not required. However the authors should strengthen the validation of their results by more comparative analyses of data and tone down some of their conclusions. Specific points to be taken into account when modifying the text are as follows:

1) The results are based on the Dam-ID approach to discover transcription factor binding sites. It is essential to compare the findings with available published ChIP-Seq data. The authors compared their DamID peaks with NKX2-5 ChIP-Seq from HL-1 cells (32) mentioning 77.8% overlap but commenting that genes associated with ChIP-Seq peaks overwhelmed DamID targets, making comparison difficult. However, since a high quality ChIP-Seq dataset provides genome-wide TF occupancy information with higher resolution, the authors should show the percentage of ChIP-Seq peaks that are overlaid for the DamID data. This will serve as a validation.

2) The enzymatic function of Dam is supposed to be local. The authors should indicate the average distance from NKE to the nearest (predicted) methylated *Dpn*I sites in the enriched DamID peaks and should discuss whether the requirement for *Dpn* sites introduces bias.

3) The experiments were performed using overexpression in cell culture and the data should also be compared with results obtained by others on NKX2-5 chromatin interactions using endogenous NKX2-5 (ex. [78]).

4) The authors understandably restrict their analysis of target sites to promoter regions and do not attempt to analyse more distant regulatory regions. It would be useful to discuss what fraction of DamID selected genomic fragments are estimated to fall within these promoter regions. This could be illustrated by comparison of promoter versus known enhancer sequences for one or two genes in their data set. Mention of comparison with the data of He et al. (in the subsection “DamID identification of NKX2-5 WT targets”) is unclear.

5) Conclusions should be toned down, with inclusion of reference to a number of caveats – overexpression of proteins in cultured cell lines, limited in vivo validation, no evidence provided that expression of off target genes is increased in mouse models or patients with NKX2-5 mutations.

A specific caveat concerns the presence of wild type NKX2-5 protein in HL-1 cells. The authors should justify their assumption that they are not pulling down mutant protein interacting with endogenous wild type protein which might mean that the binding effects are secondary to the binding of wild type NKX2-5. In the context of the ES cell experiments, it is not clear why complex formation with the endogenous NKX2-5 protein does not permit more nuclear translocation (in the first paragraph of the subsection headed “DamID identification of NKX2-5 WT targets”).

6) What about the activity of NKX2-5 as a repressor, previously demonstrated by the Harvey lab and also for Tinman in *Drosophila*. Immunofluorescence shown for NKX2-5 (Figure 1—figure supplement 8) only co-localises with active histone marks in HL-1 cells. What about its effect on *Id3* transcription? The comments on activation/repression in the context of the ES cell experiments (in the subsection headed “Functionality of NKX2-5 mutants”) require more explanation. The authors should discuss their thoughts about the potential mechanism in repressive functions of NKX2-5.

7) Results are mentioned for NF1-B1/3 (in the fourth paragraph of the subsection headed “DamID identification of NKX2-5 WT targets”) but the in vivo significance of this is not clear and the factors are not integrated into the gene regulatory network (Figure 7). Is it necessary to include this?

8) It is not clear that the ELK complex is playing a highly important role. It only came up towards the bottom of the motif enrichment shown in Figure 2—figure supplement 2. The authors should explain why they chose to concentrate on the association of this complex with NKX2-5. The authors claim that the GO analysis shows that SRF and ELK1/4 regulate many cellular functions. This may be the case, although target genes appear to be expressed at a low level in the heart, suggesting that the major impact of these factors may be in the hijacking phenomenon? In the absence of complete transcriptome studies and functional promoter analysis statements about the importance of the role of the ELK complex should be qualified (see also point 4).

Concerning the role of ETS factors in cardiogenesis, the highly relevant paper of Woznica et al. (Dev Biol 2012) in the *Ciona* model should be discussed. In addition to showing the importance of ETS factors in the *Ciona* cardiac gene regulatory network, this study found an adjacent ATTA motif similar to the over-represented motif identified by Bouveret et al.

---

## [Author Response]

*1) The results are based on the Dam-ID approach to discover transcription factor binding sites. It is essential to compare the findings with available published ChIP-Seq data. The authors compared their DamID peaks with NKX2-5 ChIP-Seq from HL-1 cells (*[32]*) mentioning 77.8% overlap but commenting that genes associated with ChIP-Seq peaks overwhelmed DamID targets, making comparison difficult. However, since a high quality ChIP-Seq dataset provides genome-wide TF occupancy information with higher resolution, the authors should show the percentage of ChIP-Seq peaks that are overlaid for the DamID data. This will serve as a validation*.

We agree that this is an important issue. In response to the reviewer’s request, we calculated the overlap of NKX2-5 DamID peaks (this study) and ChIP-seq peaks (32), both data sets generated in HL-1 cells. A proportional Venn diagram is now included in a new figure (Figure 1—figure supplement 9). We also included in this figure a comparison of NKX2-5 ChIP-seq peaks generated from total adult hearts by van den Boogaard and colleagues (78), as requested below (point 3). Figure legends and the Materials and methods section have been adjusted accordingly. Since our DamID peaks were detected on Affymetrix promoter microarrays (average -6kb to +2.5kb relative to TSS), we limited the comparison to genomic regions covered by the microarray.

The overlap between our DamID data and the ChIP-seq data of He et al. data, both generated in HL-1 cells, was relatively low (18% of DamID peaks and 6.8% of ChIP-seq peaks). Importantly, the overlap between the He et al. peaks and those of van den Boogaard peaks, both established using ChIP-seq, was similarly low (15.6% of the van en Boogaard et al. peaks and 7.6% of the He et al. peaks).

We note the following observations that relate to these specific comparisons of NKX2-5 targets:

A) Previous studies comparing DamID and ChIP-chip experiments performed in *Drosophila* reported “a high degree of overlap” (60, 57, 73, 76, 90). Although we did not compare DamID with ChIP-chip directly, 10/11 DamID targets were confirmed using ChIP-PCR.

B) Contrary to the understanding of one of the reviewers (point 3 and response below), in DamID experiments extremely low, undetectable, levels of NKX2-5 were expressed from the uninduced heat shock protein68 promoter (*hsp68*). The He et al. data was generated from stable HL-1 cell lines over-expressing *bio*-tagged NKX2-5 protein (4-5x based on RNA detection). As highlighted by the editor, overexpression may lead to a larger number of apparent target peaks and we note that in the He et al. study, >20,000 peaks were detected.

C) We performed DamID using 3-4 biological replicates for each experiment. It is known that ChIP-seq experiments can be noisy ([Bibr bib89a]); therefore replication is an essential design element for ChIP-seq experiments. The ChIP-seq experiments of He et al. and van den Boogaard et al. were performed with no biological or technical replicates.

D) Within our total DamID peaks, de novo motif discovery identified the perfect known NKE (T/CAAGTG) as the top enriched motif, with at least one perfect NKE detected in 79% of peaks and 2 or more perfect NKEs detected in 40% of peaks. Up to 12 such NKEs were detected in NKX2-5 peaks. NKX2-5 ChIP-seq peaks determined by He et al. and van den Boogaard et al. were most enriched in a variant form of that NKE (GAGT/AG) which was determined from a limited selection of either the top 500 peaks or 600 randomly-selected peaks, respectively. This variant has not been documented previously using techniques such as SELEX or protein-binding microarrays. We note from our calculations that the high affinity NKE (AAGTG) was present in 59% and 43%, respectively, of their ChIPseq peaks included in Figure 1—figure supplement 9, significantly less than in our DamID peak set (79%).

E) Western blots presented in Figure S1D (32) show that *bio*-tagged NKX2-5 fusions both before and after enrichment by immunoprecipitation were highly degraded. ChIP-seq data presented in this paper therefore likely represents targets of heterodisperse NKX2-5 processed products.

F) NKX2-5 target genes identified by DamID were most enriched in cardiac GO terms with very low p-values. When restricted to promoter regions, NKX2-5 ChIP-seq peaks were not enriched in any GO terms containing “cardiac”, “muscle” or “heart” (32) or were moderately enriched in cardiac-related terms with higher p-values (78).

We strongly believe that our DamID data should be evaluated on its own merits. There are many findings in our DamID data and in previously published DamID data that suggest that this technique is highly specific. As we have highlighted above, there are potentially simple and well-documented experimental design explanations (e.g. inclusion of replicates) for the differences we have observed. Although we agree that the issues facing the field in this regard are important and warrant further investigation, we prefer to deal with them briefly in our paper and to follow up with a subsequent meta-analysis of comparable experimental designs.

For the present paper, we present the overlap of DamID peak with those of He et al. and van den Boogaard et al., as requested, and provide a brief commentary in the second paragraph of the Discussion.

*2) The enzymatic function of Dam is supposed to be local. The authors should indicate the average distance from NKE to the nearest (predicted) methylated* Dpn*I sites in the enriched DamID peaks and should discuss whether the requirement for* Dpn *sites introduces bias*.

It is correct to say that DamID peaks, in particular their length, can be influenced by the density of *Dpn*I sites in proximity of binding sites, which occur on average every 260bp in the mouse genome. In our experiments on HL-1 cells, *Dpn*I digestion and amplification lead to fragments of ∼200-2000bp (see Figure 1—figure supplement 3). We also calculated, that for all perfect NKX2-5 high affinity binding sites (NKE; 5’AAGTG3’) in the mouse genome, 90% had a *Dpn*I site within 1,000bps upstream or downstream of the NKE, suggesting that DamID captures the vast majority of NKX2-5 direct targets.

We have added a comment about the length of peaks and our calculation of bias as discussed above in the text (in the subsection headed “DamID identification of NKX2-5 WT target”). We have also added a new track in Figure 1—figure supplement 7 showing the *Dpn*I sites.

*3) The experiments were performed using overexpression in cell culture and the data should also be compared with results obtained by others on NKX2-5 chromatin interactions using endogenous NKX2-5 (ex.*
[78]*)*.

The statement above about overexpression is incorrect. We expressed DamID fusions in HL-1 cells at extremely low levels (undetectable by western blotting) from an uninduced *hsp68* (heat shock) promoter to avoid skewing of the network. Such low levels of expression are nonetheless sufficient for specific methylation of chromatin. Please note that overexpression shown in Figure 1—figure supplement 1 in HEK 293 cells was induced upon induction with Ponasterone A only to verify that Dam fusions were stable.

We have clarified this point in the text (in the subsection headed “DamID identification of NKX2-5 WT targets”).

In response to the second part of Question 2, we have compared NKX2-5 DamID peaks with peaks determined using ChIP-seq in the adult mouse heart (78) (Figure 1—figure supplement 9). See our response to Question 1 above.

*4) The authors understandably restrict their analysis of target sites to promoter regions and do not attempt to analyse more distant regulatory regions. It would be useful to discuss what fraction of DamID selected genomic fragments are estimated to fall within these promoter regions. This could be illustrated by comparison of promoter versus known enhancer sequences for one or two genes in their data set. Mention of comparison with the data of He et al. (in the subsection “DamID identification of NKX2-5 WT targets”) is unclear*.

The Affymetrix promoter microarray covers an estimated 7.3% of the mouse genome. This microarray would capture 20% to 30% of NKX2-5 peaks previously determined using ChIP-seq by He et al. and van den Boogaard et al., respectively.

This was noted in the third paragraph of the subsection headed “DamID identification of NKX2-5 WT targets”. We have also clarified the text in the second paragraph of the Discussion with respect to the comparison of DamID peaks and ChIPseq peaks of He et al.

*5) Conclusions should be toned down, with inclusion of reference to a number of caveats – overexpression of proteins in cultured cell lines, limited in vivo validation, no evidence provided that expression of off target genes is increased in mouse models or patients with NKX2-5 mutations*.

We confirm again for clarity that DamID fusions were expressed in HL-1 cells at extremely low levels (undetectable by western blotting). See our response to Question 3 above.

We have been generally more cautious about our claims and have included the caveats as suggested above at the end of the Discussion.

*A specific caveat concerns the presence of wild type NKX2-5 protein in HL-1 cells. The authors should justify their assumption that they are not pulling down mutant protein interacting with endogenous wild type protein which might mean that the binding effects are secondary to the binding of wild type NKX2-5*.

Response: DamID experiments do not involve “pulling down” mutant protein. NKX2-5 WT and mutant Dam fusions are expressed individually at very low levels in HL-1 cells, which express endogenous NKX2-5 protein. We make very clear in the text that expressed NKX2-5 WT and mutant fusions bind to chromatin via protein:protein interaction with endogenous NKX2-5 or with other cofactors. Indeed, the binding of NKX2-5 mutants that have a crippled or missing homeodomain to genuine NKX2-5 targets *must* occur indirectly. We provide evidence and a mechanism for how this occurs.

*In the context of the ES cell experiments, it is not clear why complex formation with the endogenous NKX2-5 protein does not permit more nuclear translocation (in the first paragraph of the subsection headed “DamID identification of NKX2-5 WT targets”)*.

In the ES cell differentiation experiments, we used the embryoid body (EB) method (see Materials and methods, “Cell lines”). Only a fraction of cells differentiate into cardiomyocytes and express endogenous NKX2-5 at day 6 of differentiation. The vector used for exogenous expression utilised a tet-responsive promoter which is likely ubiquitous in tissue competence. Thus, after application of doxycycline, exogenous NKX2-5 is induced in all cells, the majority of which do not express NKX2-5 or cardiac cofactors and therefore do not localize NKX2-5ΔHD to the nucleus. It was therefore not possible to compare results with NKX2-5ΔHD directly to those using nuclear NKX2-5 proteins (WT or Y191C).

We have modified the text in the second paragraph of the subsection headed “Functionality of NKX2-5 mutants” to make clear that endogenous NKX2-5 is only activated in a minority of cells at day 6 of differentiation, while exogenous NKX2-5 is activated in most or all cells.

*6) What about the activity of NKX2-5 as a repressor, previously demonstrated by the Harvey lab and also for Tinman in* Drosophila*. Immunofluorescence shown for NKX2-5 (*Figure 1—figure supplement 8*) only co-localises with active histone marks in HL-1 cells. What about its effect on* Id3 *transcription? The comments on activation/repression in the context of the ES cell experiments (in the subsection headed “Functionality of NKX2-5 mutants”) require more explanation. The authors should discuss their thoughts about the potential mechanism in repressive functions of NKX2-5*.

The role of NKX2-5 as a repressor was discovered in the context of early cardiac specification in the embryo (85, 63). In *NKX2-5* mutant embryos, the majority of differentially expressed genes were down-regulated (suggesting that these were positively-regulated targets of NKX2-5) although a limited number of progenitor genes were up-regulated in the transition from progenitor state to differentiated cardiomyocytes, suggesting repression of some targets. In Watanabe et al., we show that repression of the *Fgf10* genes is direct. Consistent with an early repressive role for NKX2-5, cardiac reprogramming of fibroblasts is stimulated by expression of MEF2C, GATA4 and TBX5 but inhibited by NKX2-5 expression (38).

These results are consistent with our observations that the majority of NKX2-5 target genes discovered by DamID are positively regulated and with the apparent co-localisation of (the majority of) NKX2-5 protein with active histone marks. However, our immunofluorescence results do not exclude a repressive role for NKX2-5 and indeed show that some NKX2-5 protein does not localise with active histone marks. Overall, it is clear that there is a repressive role for NKX2-5 on some genes, specifically progenitor genes, including *Id3*.

We have modified the text in the subsection headed “Probing the mechanisms of CHD using DamID” to make these issues clearer.

*7) Results are mentioned for NF1-B1/3 (in the fourth paragraph of the subsection headed “DamID identification of NKX2-5 WT targets”) but the in vivo significance of this is not clear and the factors are not integrated into the gene regulatory network (*Figure 7*). Is it necessary to include this?*

The co-enrichment of NKX2-5 and NF1 motifs and the physical interaction between NKX2-5 and NF1-B1/3 suggests that these factors co-operate to regulate NKX2-5 targets. However, we have not yet investigated this relationship further. This will be the subject of further studies. We include these data here because the NF1 motif is the third most overrepresented DNA motif in NKX2-5 peaks, after the known NKX2-5 motif (NKE) and the GATA4 motifs.

*8) It is not clear that the ELK complex is playing a highly important role. It only came up towards the bottom of the motif enrichment shown in*
Figure 2—figure supplement 2*. The authors should explain why they chose to concentrate on the association of this complex with NKX2-5. The authors claim that the GO analysis shows that SRF and ELK1/4 regulate many cellular functions. This may be the case, although target genes appear to be expressed at a low level in the heart, suggesting that the major impact of these factors may be in the hijacking phenomenon? In the absence of complete transcriptome studies and functional promoter analysis statements about the importance of the role of the ELK complex should be qualified (see also point 4)*.

We chose to focus on ETS transcription factors because the known ETS motif was significantly enriched in NKX2-5∆HD C-set peaks compared to random peaks and occurred most frequently (in 42% of NKX2-5∆HD C-set peaks) (Figure 2—figure supplement 2). We specifically chose ELK factors since they have been indirectly associated with cardiogenesis previously (see point below). Overall, the GO analysis of SRF and ELK1/4 targets showed generic cellular functions such as cytoskeletal organization and RNA processing, with the level of expression of these genes in heart being low. The point is that these ancient TFs are broadly or ubiquitously expressed and are not wholly dedicated to the cardiac-specific functions. However, they have been co-opted to regulate certain cardiac genes. It is our hypothesis that ubiquitous factors of this sort provide vital signalling inputs into the cardiac network. Furthermore, we have demonstrated that ELK1/4 can guide NKX2-5∆HD to off-targets that include the non-cardiac targets of ELK factors.

We clarify the occurrence and enrichment (relative to random peaks) of the ETS motifs in the second paragraph of the subsection headed “ETS factor binding sites are enriched in NKX2-5∆HD off-targets”.

*Concerning the role of ETS factors in cardiogenesis, the highly relevant paper of Woznica et al. (Dev Biol 2012) in the* Ciona *model should be discussed. In addition to showing the importance of ETS factors in the* Ciona *cardiac gene regulatory network, this study found an adjacent ATTA motif similar to the over-represented motif identified by Bouveret et al*.

We agree that studies in *Ciona* heart development focusing on the role of Ets/2 are relevant.

We include this in the fifth paragraph of our Discussion. We have also added the relevant references (17; 89) here and in the subsection headed “ETS factor binding sites are enriched in NKX2-5∆HD off-targets”.